# *Don't Overthink it.* PREFERRING SHORTER THINKING CHAINS FOR IMPROVED LLM REASONING

## ABSTRACT

Reasoning large language models (LLMs) heavily rely on scaling test-time compute to perform complex reasoning tasks by generating extensive "thinking" chains. While demonstrating impressive results, this approach incurs significant computational costs and inference time. In this work, we challenge the assumption that long thinking chains results in better reasoning capabilities. We first demonstrate that shorter reasoning chains within individual questions are significantly more likely to yield correct answers—up to $34.5\%$ more accurate than the longest chain sampled for the same question. Based on these results, we suggest *short-m@k*, a novel reasoning LLM inference method. Our method executes $k$ independent generations in parallel and halts computation once the first $m$ thinking processes are done. The final answer is chosen using majority voting among these $m$ chains. Basic *short-1@k* demonstrates similar or even superior performance over standard majority voting in low-compute settings—using up to $40\%$ fewer thinking tokens. *short-3@k*, while slightly less efficient than *short-1@k*, consistently surpasses majority voting across all compute budgets, while still being substantially faster (up to $33\%$ wall time reduction). To further validate our findings, we finetune an LLM using short, long, and randomly selected reasoning chains. We then observe that training on the shorter ones leads to better performance. Our findings suggest rethinking current methods of test-time compute in reasoning LLMs, emphasizing that longer "thinking" does not necessarily translate to improved performance and can, counter-intuitively, lead to degraded results.

## 1 INTRODUCTION

Scaling test-time compute has been shown to be an effective strategy for improving the performance of reasoning LLMs on complex reasoning tasks (OpenAI, 2024; 2025; Team, 2025b). This method involves generating extensive *thinking*—very long sequences of tokens that contain enhanced reasoning trajectories, ultimately yielding more accurate solutions. Prior work has argued that longer model responses result in enhanced reasoning capabilities (Guo et al., 2025; Muennighoff et al., 2025; Anthropic, 2025). However, generating such long-sequences also leads to high computational cost and slow decoding time due to the autoregressive nature of LLMs.

In this work, we demonstrate that scaling test-time compute does not necessarily improve model performance as previously thought. We start with a somewhat surprising observation. We take four leading reasoning LLMs, and for each generate multiple answers to each question in four complex reasoning benchmarks. We then observe that taking the *shortest* answer for each question strongly and consistently outperforms both a strategy that selects a random answer (up to $18.8\%$ gap) and one that selects the longest answer (up to $34.5\%$ gap). These performance gaps are on top of the natural reduction in sequence length—the shortest chains are $50\%$ and $67\%$ shorter than the random and longest chains, respectively.

Building on these findings, we propose *short-m@k*—a novel inference method for reasoning LLMs. *short-m@k* executes $k$ generations in parallel and terminates computation for all generations as soon as the first $m$ thinking processes are completed. The final answer is then selected via majority voting among those shortest chains, where ties are broken by taking the shortest answer among the tied candidates. See Figure 1 for visualization.

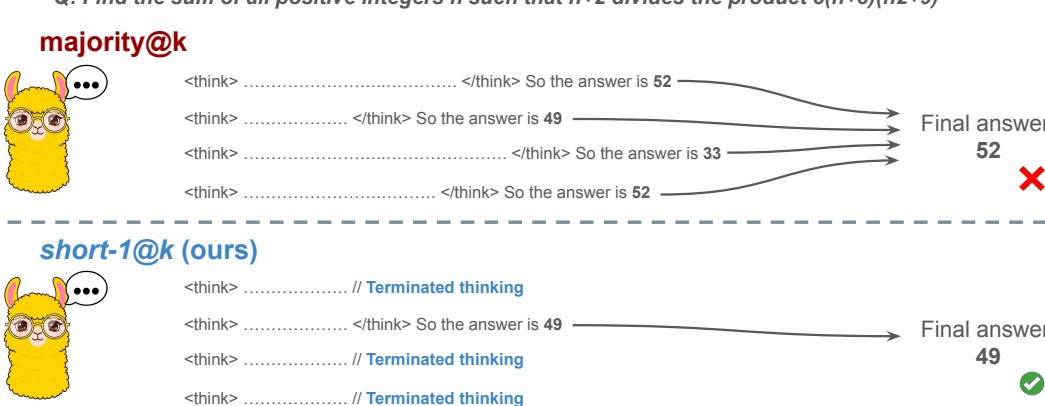

Figure 1: Visual comparison between majority voting and our proposed method *short-m@k* with $m = 1$ ("..." represent thinking time). Given $k$ parallel attempts for the same question, majority@$k$ waits until all attempts are done, and perform majority voting among them. On the other hand, our *short-m@k* method halts computation for all attempts as soon as the first $m$ attempts finish "thinking", which saves much compute and time, and surprisingly also boost performance in most cases.

We evaluate *short-m@k* using four reasoning LLMs, and compare it to majority voting—the most common aggregation method for evaluating reasoning LLMs on complex benchmarks (Wang et al., 2022; Abdin et al., 2025). We show that in low-compute regimes, *short-1@k*, i.e., taking the *single* shortest chain, outperforms majority voting, while significantly reducing the time and compute needed to generate the final answer. For example, using LN-Super-49B (Bercovich et al., 2025), *short-1@k* can reduce up to $40\%$ of the compute while giving the same performance as majority voting. Moreover, for high-compute regimes, *short-3@k*, which halts generation after three thinking chains are completed, consistently outperforms majority voting across all compute budgets, while running up to $33\%$ faster.

To gain further insights into the underlining mechanism of shorter thinking is preferable, we analyze the generated reasoning chains. We first show that while taking the shorter reasoning is beneficial per individual question, longer reasoning is still needed to solve harder questions, as claimed in recent studies (Anthropic, 2025; OpenAI, 2024; Muennighoff et al., 2025). Next, we analyze the backtracking and re-thinking behaviors of reasoning chains. We find that shorter reasoning paths are more effective, as they involve fewer backtracks, with a longer average backtrack length. This finding holds both generally and when controlling for overall trajectory length.

To further strengthening our findings, we study whether training on short reasoning chains can lead to more accurate models. To do so, we finetune the Qwen-2.5-32B model (Yang et al., 2024) on three variants of the S1 dataset (Muennighoff et al., 2025): S1-short, S1-long, and S1-random, consisting of examples with the shortest, longest, and randomly sampled reasoning trajectories among several generations, respectively. Our experiments demonstrate that finetuning using S1-short not only yields shorter thinking lengths, but also improves model performance. Conversely, finetuning on S1-long increases reasoning time with no significant performance gains.

This work rethinks the test-time compute paradigm for reasoning LLMs, showing that longer thinking not only does not ensure better reasoning, but also leads to worse reasoning in most cases. Our *short-m@k* methods prioritize shorter reasoning, yielding improved performance and reduced computational costs for current reasoning LLMs. We also show that training reasoning LLMs with shorter reasoning trajectories can enhance performance and reduce costs. Our results pave the way towards a new era of efficient and high-performing reasoning LLMs.

## 2 RELATED WORK

**Reasoning LLMs and test-time scaling.** Reasoning LLMs tackle complex tasks by employing extensive reasoning processes, often involving detailed, step-by-step trajectories (OpenAI, 2024;

2025; Team, 2025b; Abdin et al., 2025; Anthropic, 2025; Bercovich et al., 2025; Guo et al., 2025; sky, 2025; DeepMind, 2025; Team, 2025a). This capability is fundamentally based on techniques like chain-of-thought (CoT; Wei et al., 2022), which encourage models to generate intermediate reasoning steps before arriving at a final answer. Modern LLMs use a large number of tokens, often referred to as "thinking tokens", to explore multiple problem-solving approaches, to employ self-reflection, and to perform verification. This thinking capability has allowed them to achieve superior performance on challenging tasks such as mathematical problem-solving and code generation (Ke et al., 2025).

The LLM thinking capability is typically achieved through post-training methods applied to a strong base model. The two primary approaches to instilling or improving this reasoning ability are using reinforcement learning (RL) (Guo et al., 2025; Team, 2025b) and supervised fine-tuning (Muennighoff et al., 2025; Ye et al., 2025). Guo et al. (2025) have demonstrated that as training progresses the model tends to generate longer thinking trajectories, which results in improved performance on complex tasks. Similarly, Anthropic (2025) and Muennighoff et al. (2025) have shown a correlation between increased average thinking length during inference and improved performance. We challenge this assumption, demonstrating that shorter sequences are more likely to yield an accurate answer.

**Efficiency in reasoning LLMs.**    While shortening the length of CoT is beneficial for non-reasoning models (Nayab et al., 2024; Kang et al., 2025), it is higly important for reasoning LLMs as they require a very large amount of tokens to perform the thinking process. As a result, recent studies tried to make the process more efficient, e.g., by using early exit techniques for reasoning trajectories (Pu et al., 2025; Yang et al., 2025), by suppressing backtracks (Wang et al., 2025a) or by training reasoning models which enable control over the thinking length (Yu et al., 2025).

Several recent works studied the relationship between reasoning trajectory length and correctness. Lu et al. (2025) proposed a method for reducing the length of thinking trajectories in reasoning training datasets. Their method employs a reasoning LLM several times over an existing trajectory in order to make it shorter. As this approach eventually trains a model over shorter trajectories it is similar to the method we employ in Section 6. However, our method is simpler as it does not require an LLM to explicitly shorten the sequence. Fatemi et al. (2025); Qi et al. (2025) and Arora & Zanette (2025) proposed RL methods to shorten reasoning in language models. Fatemi et al. (2025) also observed that correct answers typically require shorter thinking trajectories by averaging lengths across examples, suggesting that lengthy responses might inherently stem from RL-based optimization during training. In Section 5.1 we show that indeed correct answers usually use shorter thinking trajectories, but also highlight that averaging across all examples might hinder this effect as easier questions require sustainably lower amount of reasoning tokens compared to harder ones.

More relevant to our work, Wu et al. (2025) showed that there is an optimal thinking length range for correct answers according to the difficulty of the question, while Wang et al. (2025b) found that for a specific question, correct responses from reasoning models are usually shorter than incorrect ones. We provide further analysis supporting these observations in Sections 3 and 5. Finally, our proposed inference method *short-m@k* is designed to enhance the efficiency of reasoning LLMs by leveraging this property, which can be seen as a generalization of the FFS method (Agarwal et al., 2025), which selects the shortest answer among several candidates as in our *short-1@k*.

## 3   SHORTER THINKING IS PREFERABLE

As mentioned above, the common wisdom in reasoning LLMs suggests that increased test-time computation enhances model performance. Specifically, it is widely assumed that longer reasoning process, which entails extensive reasoning thinking chains, correlates with improved task performance (OpenAI, 2024; Anthropic, 2025; Muennighoff et al., 2025). We challenge this assumption and ask whether generating more tokens per trajectory actually leads to better performance. To that end, we generate multiple answers per question and compare performance based solely on the shortest, longest and randomly sampled thinking chains among the generated samples.

### 3.1   EXPERIMENTAL DETAILS

We consider four leading, high-performing, open, reasoning LLMs. Llama-3.3-Nemotron-Super-49B-v1 [LN-Super-49B; Bercovich et al., 2025]: a reasoning RL-enhanced version of Llama-3.3-

Table 1: Shorter thinking performs better. Comparison between taking the shortest/longest/random generation per example.

| | GPQA-D | | AIME 2024 | | AIME 2025 | | HMMT | | Math Average | |
|---|---|---|---|---|---|---|---|---|---|---|
| | Thinking Tokens ↓ | Acc. ↑ | Thinking Tokens ↓ | Acc. ↑ | Thinking Tokens ↓ | Acc. ↑ | Thinking Tokens ↓ | Acc. ↑ | Thinking Tokens ↓ | Acc. ↑ |
| **LN-Super-49B** | | | | | | | | | | |
| random | 5357 | 65.1 | 11258 | 58.8 | 12105 | 51.3 | 13445 | 33.0 | 12270 | 47.7 |
| longest | 8763 (+64%) | 57.6 | 18566 | 33.3 | 18937 | 30.0 | 19790 | 23.3 | 19098 (+56%) | 28.9 |
| shortest | 2790 (−48%) | **69.1** | 6276 | **76.7** | 7036 | **66.7** | 7938 | **46.7** | 7083 (−42%) | **63.4** |
| **R1-32B** | | | | | | | | | | |
| random | 5851 | 62.5 | 9614 | 71.8 | 11558 | 56.4 | 12482 | **38.3** | 11218 | 55.5 |
| longest | 9601 (+64%) | 57.1 | 17689 | 53.3 | 19883 | 36.7 | 20126 | 23.3 | 19233 (+71%) | 37.8 |
| shortest | 3245 (−45%) | **64.7** | 4562 | **80.0** | 6253 | **63.3** | 6557 | 36.7 | 5791 (−48%) | **60.0** |
| **QwQ-32B** | | | | | | | | | | |
| random | 8532 | 63.7 | 13093 | 82.0 | 14495 | **72.3** | 16466 | 52.5 | 14685 | 68.9 |
| longest | 12881 (+51%) | 54.5 | 20059 | 70.0 | 21278 | 63.3 | 24265 | 36.7 | 21867 (+49%) | 56.7 |
| shortest | 5173 (−39%) | **64.7** | 8655 | **86.7** | 10303 | 66.7 | 11370 | **60.0** | 10109 (−31%) | **71.1** |
| **R1-670B** | | | | | | | | | | |
| random | 11843 | **76.2** | 16862 | **83.8** | 18557 | 82.5 | 21444 | 68.2 | 18954 | 78.2 |
| longest | 17963 (+52%) | 63.1 | 22603 | 70.0 | 23570 | 66.7 | 27670 | 40.0 | 24615 (+30%) | 58.9 |
| shortest | 8116 (−31%) | 75.8 | 11229 | 83.3 | 13244 | **83.3** | 13777 | **83.3** | 12750 (−33%) | **83.3** |

70B (Grattafiori et al., 2024); R1-Distill-Qwen-32B [R1-32B; Guo et al., 2025]: an SFT finetuned version of Qwen-2.5-32B-Instruct (Yang et al., 2024) derived from R1 trajectories; QwQ-32B a reasoning RL-enhanced version Qwen-2.5-32B-Instruct (Team, 2025b); and R1-0528 a 670B RL-trained flagship reasoning model (R1-670B; Guo et al., 2025).

We evaluate all models using four competitive reasoning benchmarks. We use AIME 2024 (of America, 2024), AIME 2025 (of America, 2025) and HMMT February 2025, from the Math Arena benchmark (Balunović et al., 2025). This three benchmarks are derived from math competitions, and involve solving problems that cover a broad range of mathematics topics. Each dataset consists of 30 examples with varied difficulty. We also evaluate the models using the GPQA-diamond benchmark [GPQA-D; Rein et al., 2024], which consists of 198 multiple-choice scientific questions, and is considered to be challenging for reasoning LLMs (DeepMind, 2025).

For each question, we generate 20 responses per model, yielding a total of about 23k generations. For all models we use temperature of 0.7, top-p=0.95 and a maximum number of generated tokens of 32, 768. When measuring the thinking chain length, we measure the token count between the `<think>` and `</think>` tokens. We run inference for all models using paged attention via the vLLM framework (Kwon et al., 2023).

## 3.2 THE SHORTER THE BETTER

For all generated answers, we compare *short* vs. *long* thinking chains for the *same question*, along with a random chain. Results are presented in Table 1.[1] First, as expected, the shortest answers are 25%–50% shorter compared to randomly sampled responses. However, we also note that across almost all models and benchmarks, considering the answer with the shortest thinking chain actually boosts performance, yielding an average absolute improvement of 2.2%–15.7% on the math benchmarks compared to randomly selected generations. When considering the longest thinking answers among the generations, we further observe an increase in thinking chain length, with up to 75% more tokens per chain. These extended reasoning trajectories substantially degrade performance, resulting in average absolute reductions ranging between 5.4%–18.8% compared to random generations over all benchmarks. These trends are most noticeable when comparing the shortest generation with the longest ones, with an absolute performance gain of up to 34.5% in average accuracy and a substantial drop in the number of thinking tokens.

The above results suggest that long generations might come with a significant price-tag, not only in running time, but also in performance. That is, within an individual example, shorter thinking

---

[1]In this section we exclude generations where thinking is not completed within the maximum generation length, as these often result in an infinite thinking loop.

trajectories are much more likely to be correct. In Section 5.1 we examine how these results relate to the common assumption that longer trajectories leads to better LLM performance. Next, we propose strategies to leverage these findings to improve the efficiency and effectiveness of reasoning LLMs.

# 4 *short-m@k*: FASTER AND BETTER INFERENCE OF REASONING LLMs

Based on the results presented in Section 3, we suggest a novel inference method for reasoning LLMs. Our method—*short-m@k*—leverages batch inference of LLMs per question, using multiple parallel decoding runs for the same query. We begin by introducing our method in Section 4.1. We then describe our evaluation methodology, which takes into account inference compute and running time (Section 4.2). Finally, we present our results (Section 4.3).

## 4.1 THE *short-m@k* METHOD

The *short-m@k* method, visualized in Figure 1, performs parallel decoding of $k$ generations for a given question, halting computation across all generations as soon as the $m \leq k$ shortest thinking trajectories are completed. It then conducts majority voting among those shortest answers, resolving ties by selecting the answer with the shortest thinking chain. Given that thinking trajectories can be computationally intensive, terminating all generations once the $m$ shortest trajectories are completed not only saves computational resources but also significantly reduces wall time due to the parallel decoding approach, as shown in Section 4.3.

Below we focus on *short-1@k* and *short-3@k*, with *short-1@k* being the most efficient variant of *short-m@k* and *short-3@k* providing the best balance of performance and efficiency (see Section 4.3). Ablation studies on $m$ and other design choices are presented in Appendix D.

## 4.2 EVALUATION SETUP

We evaluate all methods under the same setup as described in Section 3.1. We report the averaged results across the math benchmarks, while the results for GPQA-D presented in Appendix B. The per benchmark resutls for the math benchmarks are in Appendix C. We report results using our method (*short-m@k*) with $m \in \{1, 3\}$. We compare the proposed method to the standard majority voting (majority@$k$), arguably the most common method for aggregating multiple outputs (Wang et al., 2022), which was recently adapted for reasoning LLMs (Guo et al., 2025; Abdin et al., 2025; Wang et al., 2025b). As an oracle, we consider pass@$k$ (Kulal et al., 2019; Chen et al., 2021), which measures the probability of including the correct solution within $k$ generated responses.

We benchmark the different methods with sample sizes of $k \in \{1, 2, ..., 10\}$, assuming standard parallel decoding setup, i.e., all samples are generated in parallel. For the oracle (pass@$k$) approach, we use the unbiased estimator presented in Chen et al. (2021), with our 20 generations per question ($n=20$). For the *short-1@k* method, we use the rank-score@$k$ metric (Hassid et al., 2024), where we sort the different generations according to thinking length. For majority@$k$ and *short-m@k* where $m > 1$, we run over all $k$-sized subsets out of the 20 generations per example.

We evaluate the different methods considering three main criteria: (a) *Sample-size* (i.e., $k$), where we compare methods while controlling for the number of generated samples; (b) *Thinking-compute*, where we measure the total number of thinking tokens used across all generations in the batch; and (c) *Time-to-answer*, which measures the wall time of running inference using each method. We note that we assume that currently served LLMs operate with a fixed processing speed in a fixed batch setting while using parallel decoding. As a result, using our method (*short-m@k*), we terminate all other generations after the first $m$ decoding thinking processes terminate. Thus, the overall thinking compute is the total number of thinking tokens for each of the $k$ generations at that point. Similarly, the overall time is that of the $m$'th shortest generation process. Conversely, for majority@$k$, the method's design necessitates waiting for all generations to complete before proceeding. Hence, we consider the compute as the total amount of thinking tokens in all generations and run time according to the longest thinking chain. As for the oracle approach, we terminate all thinking trajectories once the shortest correct one is finished, and consider the compute and time accordingly.

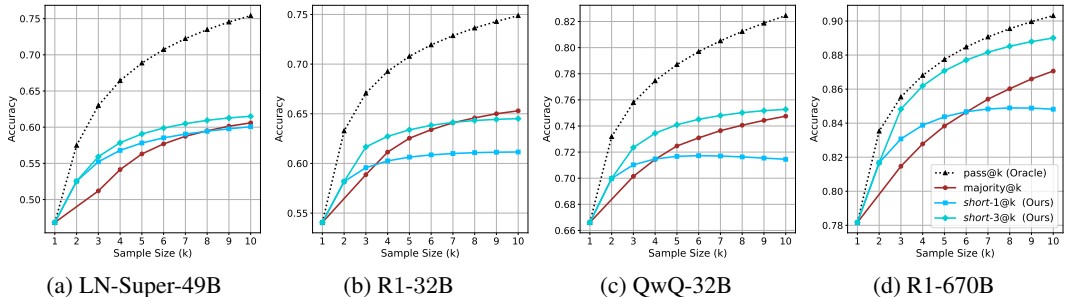

(a) LN-Super-49B      (b) R1-32B      (c) QwQ-32B      (d) R1-670B

Figure 2: Comparing different inference methods under controlled sample size ($k$). All methods improve with larger sample sizes. Interestingly, this trend also holds for the *short-m@k* methods.

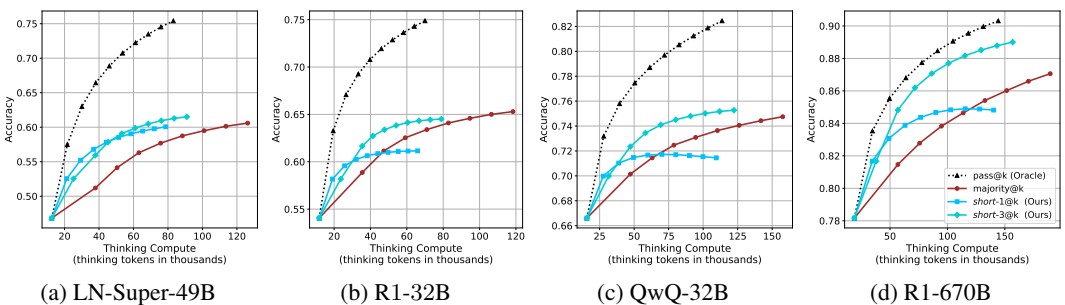

(a) LN-Super-49B      (b) R1-32B      (c) QwQ-32B      (d) R1-670B

Figure 3: Comparing different inference methods under controlled thinking compute. *short-1@k* is highly performant in low compute regimes. *short-3@k* dominates the curve compared to majority@$k$.

## 4.3 RESULTS

**Sample-size ($k$).** We start by examining different methods across benchmarks for a fixed sample size $k$. Results aggregated across math benchmarks are presented in Figure 2, while Figure 5 in Appendix B presents GPQA-D results, and detailed results per benchmark can be seen at Appendix C. We observe that, generally, all methods improve with larger sample sizes, indicating that increased generations per question enhance performance. This trend is somewhat expected for the oracle (pass@$k$) and majority@$k$ methods but surprising for our method, as it means that even when a large amount of generations is used, the shorter thinking ones are more likely to be correct. The only exception is QwQ-32B (Figure 2c), which shows a small of decline when considering larger sample sizes with the *short-1@k* method.

When comparing *short-1@k* to majority@$k$, the former outperforms at smaller sample sizes, but is outperformed by the latter in three out of four models when the sample size increases. Meanwhile, the *short-3@k* method demonstrates superior performance, dominating across nearly all models and sample sizes. Notably, for the R1-670B model, *short-3@k* exhibits performance nearly on par with the oracle across all sample sizes. We next analyze how this performance advantage translates into efficiency benefits.

**Thinking-compute.** The aggregated performance results for math benchmarks, evaluated with respect to thinking compute, are presented in Figure 3 (per-benchmark results provided in Appendix C), while the GPQA-D respective results are presented in Figure 6 in Appendix B. We again observe that the *short-1@k* method outperforms majority@$k$ at lower compute budgets. Notably, for LN-Super-49B (Figure 3a), the *short-1@k* method surpasses majority@$k$ across all compute budgets. For instance, *short-1@k* achieves 57% accuracy with approximately 60% of the compute budget used by majority@$k$ to achieve the same accuracy. For R1-32B, QwQ-32B and R1-670B models, the *short-1@k* method exceeds majority@$k$ up to compute budgets of 45k, 60k and 100k total thinking tokens, respectively, but is underperformed by it on larger compute budgets.

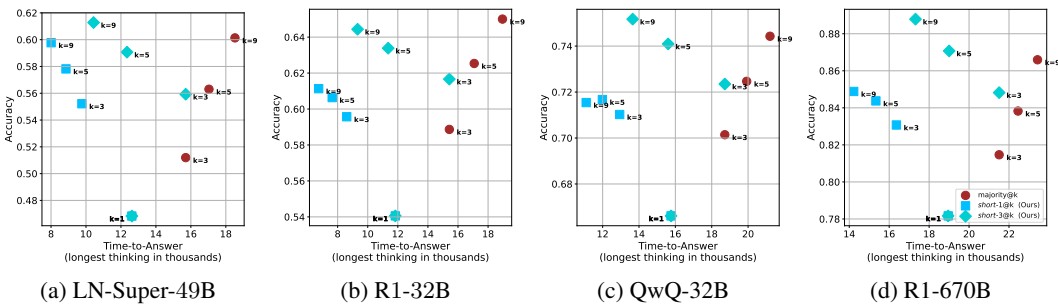

(a) LN-Super-49B    (b) R1-32B    (c) QwQ-32B    (d) R1-670B

Figure 4: Comparing time-to-answer for different inference methods. Our methods substantially reduce time cost with no major loss in performance. Unlike majority@$k$, which becomes slower as $k$ grows, our methods run **faster** with $k$, as the probability of finding a short chain increases with $k$.

The *short-3@k* method yields even greater performance improvements, incurring only a modest increase in thinking compute compared to *short-1@k*. When compared to majority@$k$, *short-3@k* consistently achieves higher performance with lower thinking compute across all models and compute budgets. For example, with the QwQ-32B model (Figure 3c), and an average compute budget of 80k thinking tokens per example, *short-3@k* improves accuracy by 2% over majority@$k$. For the R1-670B model (Figure 3d), *short-3@k* consistently outperforms majority voting, yielding an approximate 4% improvement with an average token budget of 100k.

**Time-to-answer.** Finally, the math aggregated time-to-answer results are shown in Figure 4, with GPQA-D results shown in Figure 7 and pe math benchmark in Appendix C.[2] As sample size increases, majority@$k$ exhibits longer time-to-answer, driven by a higher probability of sampling generations with extended thinking chains, requiring all trajectories to complete. Conversely, the *short-1@k* method shows *reduced* time-to-answer with larger sample sizes, as the probability of encountering a short answer increases. This trend also holds for the *short-3@k* method after three reasoning processes complete.

This phenomenon makes the *short-1@k* and *short-3@k* methods substantially more usable compared to basic majority@$k$. For example, when using the LN-Super-49B model (Figure 4a), with a sample size of 5, the *short-1@k* method reduces time consumption by almost 50% while also increasing performance by about 1.5% compared to majority@$k$. When considering a larger sample size of 9, the performance values are almost the same but *short-1@k* is more than 55% faster.

Finally, we observe that for most models and sample sizes, *short-3@k* boosts performance, while for larger ones it also reduces time-to-answer significantly. For example, on R1-32B (Figure 4b), with $k = 5$, *short-3@k* is 33% faster than majority@$k$, while reaching superior performance. A similar boost in time-to-answer and performance is observed with QwQ-32B/R1-670B and sample size 9 (Figure 4c and Figure 4d).

## 5 ANALYSIS

To obtain deeper insights into the underlying process, making shorter thinking trajectories preferable, we conduct additional analysis. We first investigate the relation between using shorter thinking per individual example, and the necessity of longer trajectories to solve more complex problems (Section 5.1). Subsequently, we analyze backtracks in thinking trajectories to better understand the characteristics of shorter trajectories (Section 5.2). In this entire section we use trajectories produced by our models as described in Section 3.1.[3]

---

[2]For readability, Figure 4 omits the oracle, and methods are compared across a subset of sample sizes.
[3]Again, we exclude generations where thinking is not completed within the generation length.

## 5.1 HARD QUESTIONS (STILL) REQUIRE MORE THINKING

We split the questions into three equal size groups according to model's success rate. Then, we calculate the average thinking length for each of the splits, and provide the average lengths for the correct and incorrect attempts per split.

Tables 2 and 5 shows the averages thinking tokens per split for the math benchmarks and GPQA-D, respectively. We first note that as observed in Section 3.2, within each question subset, correct answers are typically *shorter* than incorrect ones. This suggests that correct answers tend to be shorter, and it holds for easier questions as well as harder ones.

Table 2: Average thinking tokens for correct (C), incorrect (IC) and all (A) answers, per split by difficulty for the math benchmarks. The numbers are in thousands of tokens.

| Model | Easy C/IC/A | Medium C/IC/A | Hard C/IC/A |
|---|---|---|---|
| LN-Super-49B | 5.3/11.1/ 5.7 | 11.4/17.1/14.6 | 12.4/16.8/16.6 |
| R1-32B | 4.9/13.7/ 5.3 | 10.9/17.3/13.3 | 14.4/15.8/15.7 |
| QwQ-32B[4] | 8.4/ – / 8.4 | 14.8/21.6/15.6 | 19.1/22.8/22.3 |
| R1-670B[4] | 13.0/ – /13.0 | 15.3/20.9/15.5 | 23.0/31.7/28.4 |

Nevertheless, we also observe that models use more tokens for more challenging questions, up to a factor of 2.9. This finding is consistent with recent studies (Anthropic, 2025; OpenAI, 2024; Muennighoff et al., 2025) indicating that using longer thinking is needed in order to solve harder questions. To summarize, harder questions require a longer thinking process compared to easier ones, but within a single question (both easy and hard), shorter thinking is preferable.

## 5.2 BACKTRACK ANALYSIS

One may hypothesize that longer thinking reflect a more extensive and less efficient search path, characterized by a higher degree of backtracking and "rethinking". In contrast, shorter trajectories indicate a more direct and efficient path, which often leads to a more accurate answer.

To this end, we track several keywords identified as indicators of re-thinking and backtracking within different trajectories.[5] We then categorize the trajectories into correct and incorrect sets, and measure the number of backtracks and their average length (quantified by keyword occurrences divided by total thinking length) for each set. We present the results for the math benchmarks and GPQA-D in Tables 3 and 6, respectively.

As our results indicate, for all models and benchmarks, correct trajectories consistently exhibit fewer backtracks compared to incorrect ones. Moreover, in almost all cases, backtracks of correct answers are longer. This may suggest that correct solutions involve less backtracking where each backtrack is longer, potentially more in-depth that leads to improved reasoning, whereas incorrect ones explores more reasoning paths that are abandoned earlier (hence tend to be shorter).

Table 3: Average number of backtracks, and their average length for correct (C), incorrect (IC) and all (A) answers in math benchmarks.

| Model | # Backtracks C/IC/A | Backtrack Len. C/IC/A |
|---|---|---|
| LN-Super-49B | 106/269/193 | 88/ 70/76 |
| R1-32B | 95/352/213 | 117/ 63/80 |
| QwQ-32B | 182/269/193 | 70/ 60/64 |
| R1-670B | 188/323/217 | 92/102/99 |

Lastly, we analyze the backtrack behavior in a length-controlled manner. Specifically, we divide trajectories into bins based on their length. Within each bin, we compare the number of backtracks between correct and incorrect trajectories. Our hypothesis is that even for trajectories of comparable length, correct trajectories would exhibit fewer backtracks, indicating a more direct path to the answer. The results over the math benchmarks and GPQA-D are presented in Appendix E. As can be seen, in almost all cases, even among trajectories of comparable length, correct ones show a lower number of backtracks. The only exception is the R1-670B model over the math benchmarks. This finding further suggests that correct trajectories are superior because they spend less time on searching for the correct answer and instead dive deeply into a smaller set of paths.

---

[4]The QwQ-32B and R1-670B models correctly answered all of their easier questions in all attempts.

[5]The keywords we used are: ['but', 'wait', 'however', 'alternatively', 'not sure', 'going back', 'backtrack', 'trace back', 'hmm', 'hmmm']

Table 4: Results for our finetuned models over the S1 variants: S1-short/long/random. The S1-short model improves performance over the other two models, while using fewer thinking tokens.

| | GPQA-D | | AIME 2024 | | AIME 2025 | | HMMT | | Math Average | |
|---|---|---|---|---|---|---|---|---|---|---|
| | Thinking Tokens ↓ | Acc. ↑ | Thinking Tokens ↓ | Acc. ↑ | Thinking Tokens ↓ | Acc. ↑ | Thinking Tokens ↓ | Acc. ↑ | Thinking Tokens ↓ | Acc. ↑ |
| S1-random | 11566 | 62.5 | 16145 | **68.8** | 17798 | 59.3 | 19243 | 40.8 | 17729 | 56.3 |
| S1-long | 12279(+6.1%) | 63.7 | 16912 | 67.3 | 17973 | 58.5 | 19397 | 42.1 | 18094 (+2.1%) | 56.0 |
| S1-short | 10845(−6.2%) | **64.8** | 15364 | 68.3 | 17195 | **60.2** | 17557 | **45.2** | 16706 (−5.8%) | **57.9** |

## 6 FINETUNING USING SHORTER TRAJECTORIES

Based on our findings, we investigate whether fine-tuning on shorter reasoning chains improves LLM reasoning accuracy. While one might intuitively expect this to be the case, given the insights from Section 3 and Section 5, this outcome is not trivial. A potential counterargument is that training on shorter trajectories could discourage the model from performing necessary backtracks (Section 5.2), thereby hindering its ability to find a correct solution. Furthermore, the benefit of using shorter trajectories for bootstrapping reasoning remains an open question.

To do so, we follow the S1 paradigm, which fine-tunes an LLM to perform reasoning using only 1,000 trajectories (Muennighoff et al., 2025). We create three versions of the S1 dataset, built from examples with the shortest, longest, and random reasoning chains among several generations.

**Data creation and finetuning setup.** To construct the three variants of S1, we generate multiple responses for each S1 question-answer pair. Specifically, for each example, we produce 10 distinct answers using the QwQ-32B model, which we select for its superior performance with respect to model size (Section 3). From these 10 responses per example, we derive three dataset variants—S1-short, S1-long, and S1-random—by selecting the shortest/longest/random response, respectively. This results in three datasets, each containing the same 1,000 queries but with distinct reasoning trajectories and answers. We then finetune the Qwen-2.5-32B-Instruct model on the three S1 variants. We further detail about the generation process, the finetuning setup and evaluation setup in Appendix F.

**Finetuning results.** Results are presented in Table 4. For GPQA-D, AIME 2025 and HMMT benchmarks, the S1-short variant achieves superior performance while using fewer thinking tokens. While performance on AIME 2024 is similar across models, S1-short still demonstrates the shortest thinking. Aggregated results across math benchmarks reveal that S1-short improves relative performance by 2.8% compared to the S1-random baseline, with a reduction of 5.8% in thinking tokens. Conversely, the S1-long model consumes more tokens than S1-random, but obtains similar performance.

These results suggest that training on shorter reasoning sequences can lead to better reasoning models that exhibit reduced computational overhead. This observation aligns with our findings in Section 3, which shows that answers with shorter thinking trajectories tend to be more accurate. We believe that developing models that reason more effectively with less computation holds significant potential.

## 7 CONCLUSION

In this work, we challenged the common assumption that increased test-time computation leads to better performance in reasoning LLMs. Through empirical analysis on four complex mathematical and reasoning benchmarks, we showed that shorter reasoning chains consistently outperform longer ones, both in accuracy and computational efficiency. Building on this insight, we introduced *short-m@k*, an inference method that prioritizes early-terminating generations. *short-1@k*, our most efficient variant, is preferred over traditional majority voting in low-compute settings. *short-3@k*, while slightly less efficient, outperforms majority voting across all compute budgets. We further investigate thinking trajectories, and find that shorter thinking usually involve less backtracks, and a more direct way to solution. To further validate our findings, we fine-tuned an LLM on short reasoning trajectories and observed improved accuracy and faster runtime, whereas training on longer chains yields diminishing returns. These findings highlight a promising direction for developing faster and more effective reasoning LLMs by embracing brevity over extended computation.

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

## A    LIMITATIONS AND BROADER IMPACT

The primary limitation of the *short-m@k* method is the reliance on batch decoding, as it requires parallel generation of multiple reasoning trajectories. This dependency might restrict its applicability in scenarios where inference memory is constrained. It should be noted that *short-m@k* can be used without batch decoding, although its efficiency gains will be lower. Additionally, while we show that finetuning on shorter reasoning chains can improve performance and efficiency, our experiments are limited to a specific model (Qwen-2.5-32B-Instruct) and dataset (S1).

In terms of broader impact, this work holds promise for enhancing the efficiency and accessibility of reasoning-LLMs by significantly lowering the required computational resources and time. By reducing these barriers, the technology could become more widely available, thereby democratizing access to advanced reasoning capabilities across a broader range of users and applications. However, as is often the case with advancements in efficiency, the decreased cost and increased scalability also carry the risk of enabling wider misuse or unintended applications of these powerful models.

## B    GPQA DIAMOND RESULTS

We present below results for the GPQA-D benchmark. Figure 5, Figure 6 and Figure 7 are the sample-size/compute/time-to-answer results for GPQA-D, respectively. Table 5 correspond to the Table 2 in Section 5.1. Table 6 and Table 8 correspond to Table 3 and Table 7 in Section 5.2, respectively.

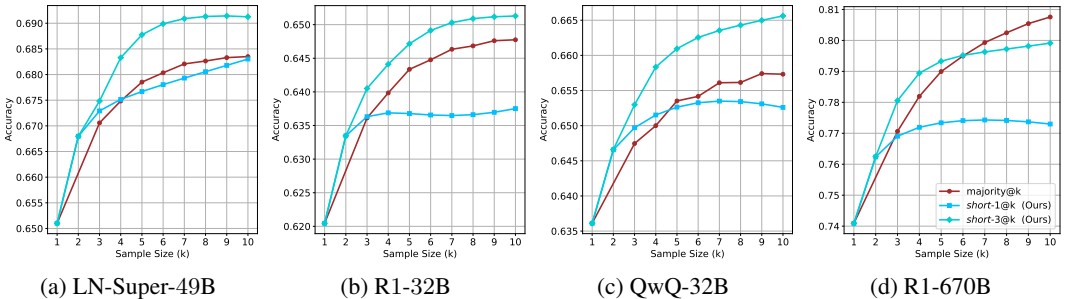

(a) LN-Super-49B          (b) R1-32B          (c) QwQ-32B          (d) R1-670B

Figure 5: GPQA-D - sample size ($k$) comparison.

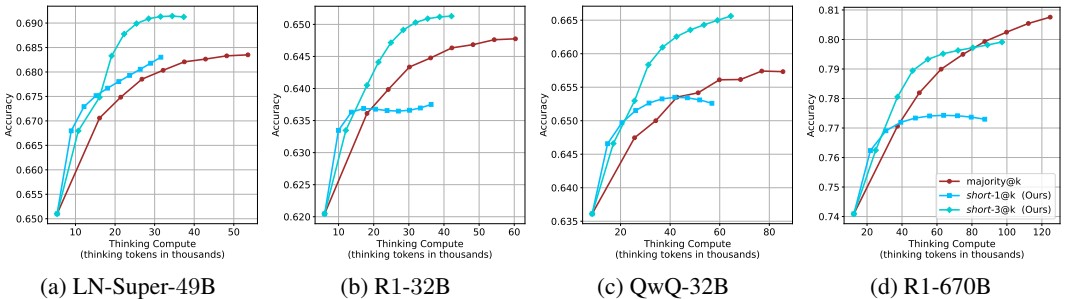

(a) LN-Super-49B          (b) R1-32B          (c) QwQ-32B          (d) R1-670B

Figure 6: GPQA-D - thinking compute comparison.

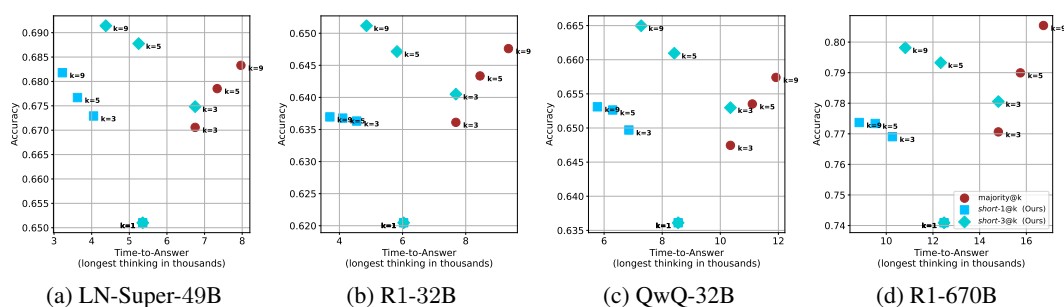

(a) LN-Super-49B  (b) R1-32B  (c) QwQ-32B  (d) R1-670B

Figure 7: GPQA-D - time-to-answer comparison.

Table 5: Average thinking tokens for correct (C), incorrect (IC) and all (A) answers, per split by difficulty for GPQA-D. The numbers are in thousands of tokens.

| Model | Easy
C/IC/A | Medium
C/IC/A | Hard
C/IC/A |
|---|---|---|---|
| LN-Super-49B | 2.5/−/2.5 | 6.2/ 7.8/ 6.6 | 7.1/ 6.9/ 7.0 |
| R1-32B | 3.4/−/3.4 | 6.4/ 7.9/ 6.8 | 8.3/ 7.8/ 7.9 |
| QwQ-32B | 5.3/−/5.3 | 8.9/13.0/ 9.7 | 11.1/10.6/10.6 |
| R1-670B | 8.1/−/8.1 | 10.9/16.0/11.4 | 17.9/17.9/17.9 |

Table 6: Average number of backtracks, and their average length for correct (C), incorrect (IC) and all (A) answers in GPQA-D.

| Model | # Backtracks
C/IC/A | Backtrack Len.
C/IC/A |
|---|---|---|
| LN-Super-49B | 89/107/ 94 | 72/56/63 |
| R1-32B | 92/173/120 | 78/48/60 |
| QwQ-32B | 152/241/178 | 52/41/46 |
| R1-670B | 122/237/156 | 83/60/69 |

## C  PER BENCHMARK RESULTS

We present the per-benchmark results for each of the criteria persented in Section 4.2. The sample-size ($k$) results are presented in Figures 8 to 10. The thinking compute comparison results are presented in Figures 11 to 13. The time-to-answer results per benchamrk are presented in Figures 14 to 16.

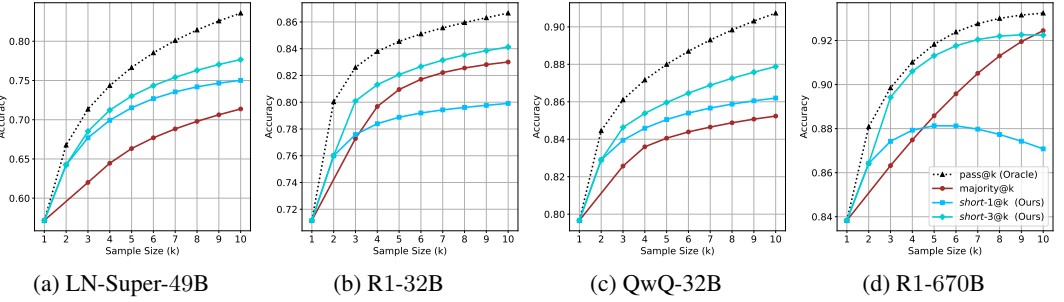

| (a) LN-Super-49B | (b) R1-32B | (c) QwQ-32B | (d) R1-670B |

Figure 8: AIME 2024 - sample size ($k$) comparison.

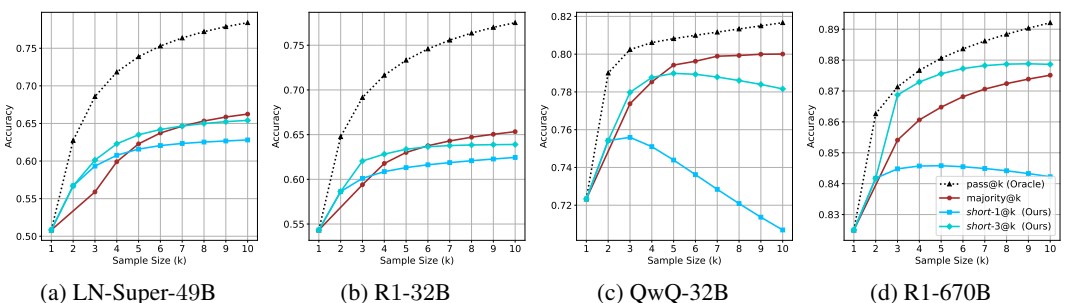

| (a) LN-Super-49B | (b) R1-32B | (c) QwQ-32B | (d) R1-670B |

Figure 9: AIME 2025 - sample size ($k$) comparison.

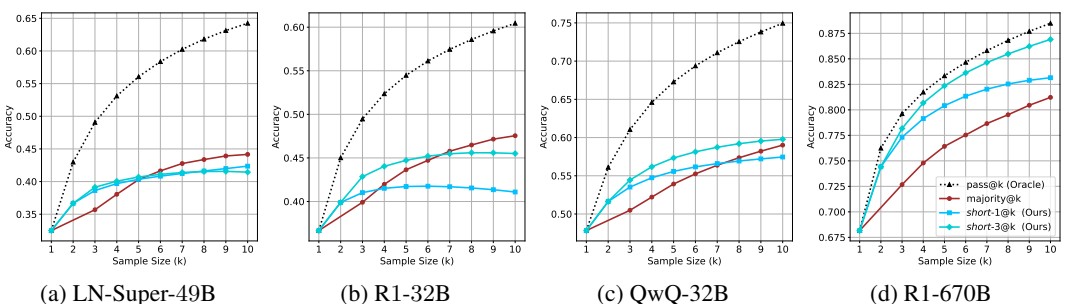

| (a) LN-Super-49B | (b) R1-32B | (c) QwQ-32B | (d) R1-670B |

Figure 10: HMMT Feb 2025 - sample size ($k$) comparison.

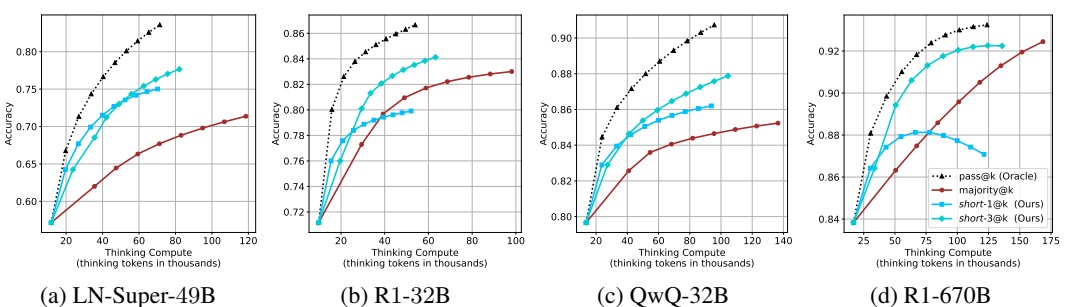

(a) LN-Super-49B    (b) R1-32B    (c) QwQ-32B    (d) R1-670B

Figure 11: AIME 2024 - thinking compute comparison.

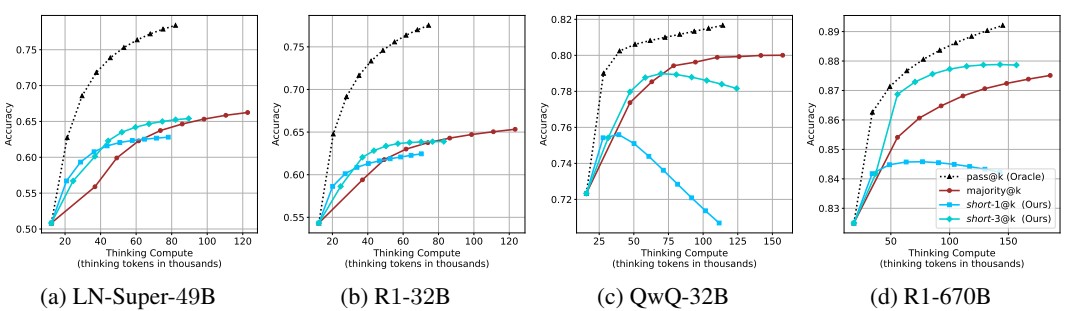

(a) LN-Super-49B    (b) R1-32B    (c) QwQ-32B    (d) R1-670B

Figure 12: AIME 2025 - thinking compute comparison.

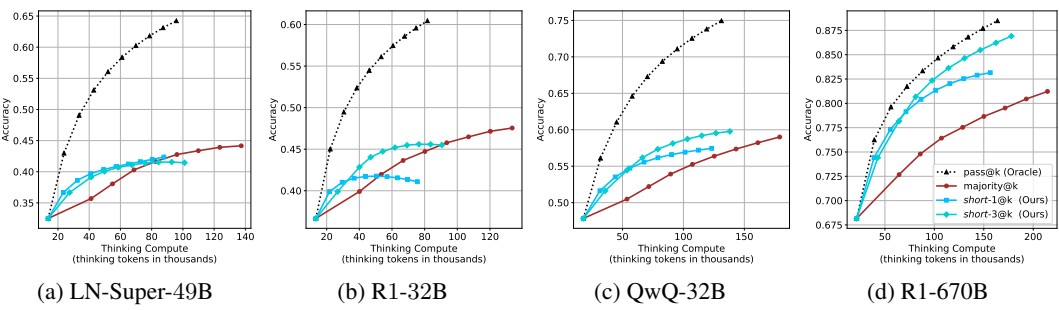

(a) LN-Super-49B    (b) R1-32B    (c) QwQ-32B    (d) R1-670B

Figure 13: HMMT Feb 2025 - thinking compute comparison.

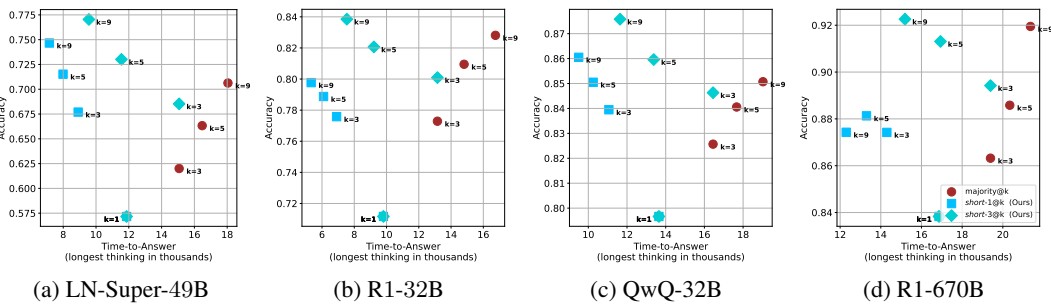

(a) LN-Super-49B    (b) R1-32B    (c) QwQ-32B    (d) R1-670B

Figure 14: AIME 2024 - time-to-answer comparison.

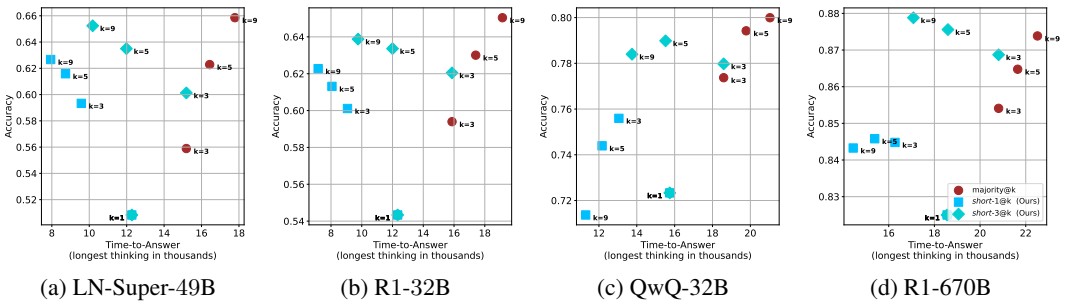

Figure 15: AIME 2025 - time-to-answer comparison.

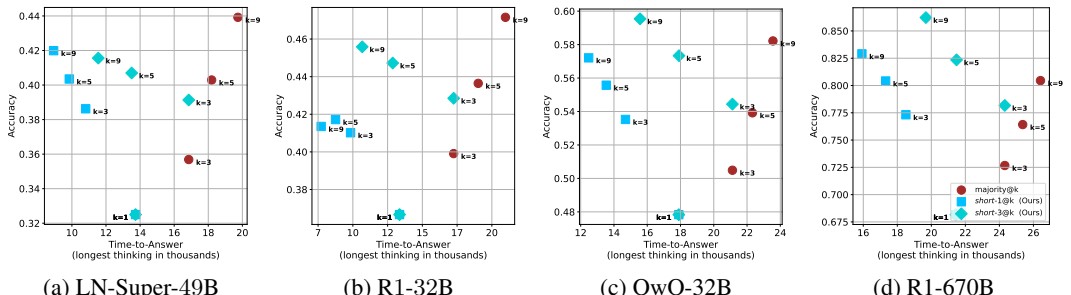

Figure 16: HMMT Feb 2025 - time-to-answer comparison.

# D ABLATION STUDIES

We investigate two axis of *short-m@k*: the value of $m$ and the tie breaking method. For all experiments we use LN-Super-49B, reporting results over the three benchmarks described in Section 3.1. For the ablation studies we focus on controlling thinking compute.

We start by ablating different $m \in \{1, 3, 4, 5, 7, 9\}$ for *short-m@k*. Results are shown in Figure 17a. As observed in our main results, *short-1@k* outperforms others in low-compute regimes, while being less effective for larger compute budgets. Larger $m$ values seem to perform similarly, with higher $m$ values yielding slightly better results in high-compute scenarios.

Next, we analyze the tie-breaking rule of *short-m@k*. We suggest the selection of the shortest reasoning chain among the vote-leading options. We compare this strategy to random tie-breaking, and to tie breaking according to the longest reasoning chain among the options. As shown in Figure 17b, the *short-m@k* strategy outperforms random tie-breaking. In contrast, choosing the option with the longest reasoning chain yields inferior results.

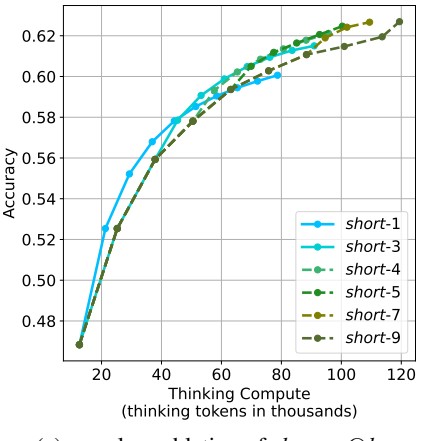

(a) $m$ values ablation of *short-m@k*

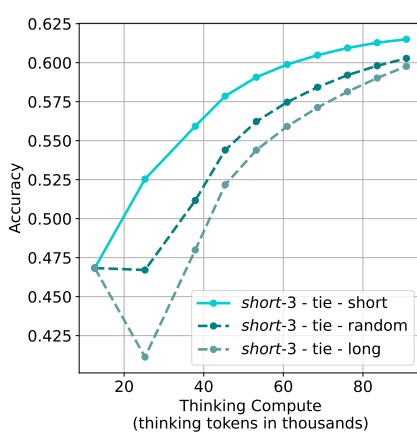

(b) Tie breaking ablation

Figure 17: Ablation studies over different $m$ values for *short-m@k*, and different tie breaking methods. Both figures show the model's average accuracy across benchmarks as a function of the length of its thinking trajectories (measured in thousands).

# E    BACKTRACKS UNDER CONTROLLED LENGTH RESULTS

Below we present the results for the backtrack count under controlled length scenarios (Section 5.2). The results over the math benchmarks are presented in Table 7 and for GPQA-D in Table 8.

Table 7: Average number of backtracks for correct (C), incorrect (IC) answers, binned by thinking length. Results are averaged across math benchmarks.

| Thinking Tokens / Model | 0-5k | 5-10k | 10-15k | 15-20k | 20-25k | 25-30k | 30-32k |
|---|---|---|---|---|---|---|---|
|  | C/IC | C/IC | C/IC | C/IC | C/IC | C/IC | C/IC |
| LN-Super-49B | 35/ 64 | 100/133 | 185/236 | 261/299 | 307/320 | 263/323 | – / 304 |
| R1-32B | 29/ 74 | 88/166 | 171/279 | 244/351 | 334/370 | 268/355 | 326/1006 |
| QwQ-32B | 50/148 | 120/174 | 194/247 | 285/353 | 354/424 | 390/476 | 551/ 469 |
| R1-670B | 58/ 27 | 100/ 86 | 143/184 | 222/203 | 264/254 | 309/289 | 352/ 337 |

Table 8: Average number of backtracks for correct (C), incorrect (IC) answers, binned by thinking length. Results are reported for GPQA-D.

| Thinking Tokens / Model | 0-5k | 5-10k | 10-15k | 15-20k | 20-25k | 25-30k | 30-32k |
|---|---|---|---|---|---|---|---|
|  | C/IC | C/IC | C/IC | C/IC | C/IC | C/IC | C/IC |
| LN-Super-49B | 38/52 | 175/164 | 207/213 | – / – | – / – | – / – | – / – |
| R1-32B | 39/54 | 194/221 | 301/375 | 525/668 | – / – | – / – | – / – |
| QwQ-32B | 65/71 | 169/178 | 333/358 | 378/544 | 357/703 | – / – | – / – |
| R1-670B | 44/72 | 93/155 | 178/232 | 297/300 | 341/341 | 463/382 | 553/477 |

## F   EXPERIMENTAL DETAILS FOR FINETUNING EXPIREMENTS

The generation process of all three variants of S1 uses the hyperparameters detailed in Section 3.1. Figure 18 shows the thinking token count histograms for the three variants of the S1 dataset (short/-long/random) presented in Section 6.

As for finetuning, we follow the S1 approach, and finetune the Qwen-2.5-32B-Instruct model on the three S1 variants. The finetuning hyperparameters are consistent with those used for the S1.1 model (Muennighoff et al., 2025), and training is conducted on 32 H100 GPUs.[6] The resulting models are evaluated using the benchmarks and experimental setup described in Section 3.1. Specifically, for each model we generate 20 answers per example, and report average accuracy.

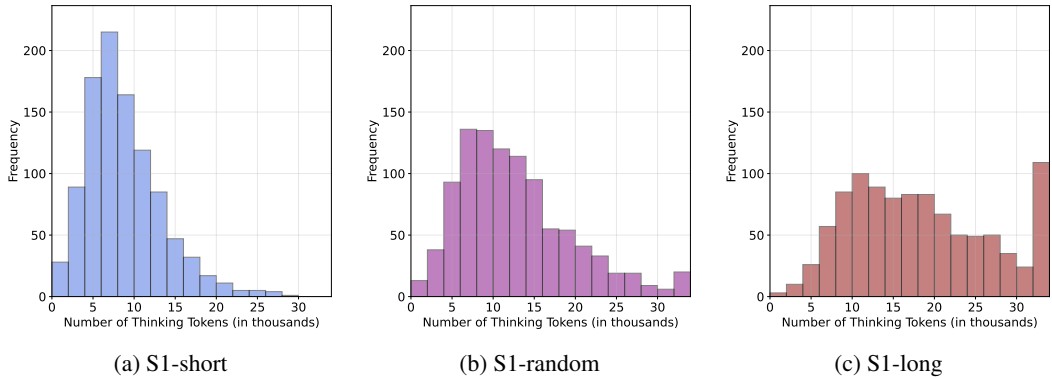

(a) S1-short                      (b) S1-random                      (c) S1-long

Figure 18: Thinking token count histograms for S1-short, S1-random and S1-long datasets.

---

[6]We match the number of gradient steps as in used for S1.1. Each model was finetuned for about 2 hours.

# G   SEQUENTIAL RESULTS

We also present our results for sequential (non-batched) decoding. To do that, we measure the amount of thinking tokens used by each method. For *short-m@k*, generation is terminated once its length exceeds the minimum length observed among the $m$ shortest previously completed generations.

The results for the math benchmarks and GPQA-D when accounting for sequential compute are presented in Figure 19 and Figure 20, respectively. While the performance of *short-m@k* shows decreased efficiency in terms of total thinking compute usage compared to a fully batched decoding setup Section 4.3, the method's superiority over standard majority voting remains. Specifically, at low compute regimes, both *short-1@k* and *short-3@k* demonstrate higher efficiency and improved performance compared to majority voting. As for most higher compute regimes, *short-m@k* outperforms the majority voting baseline.

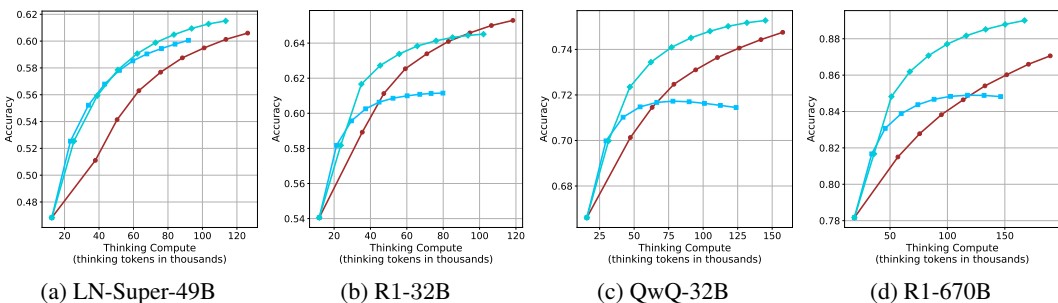

(a) LN-Super-49B    (b) R1-32B    (c) QwQ-32B    (d) R1-670B

Figure 19: Comparing different methods for the math benchmarks under sequential compute.

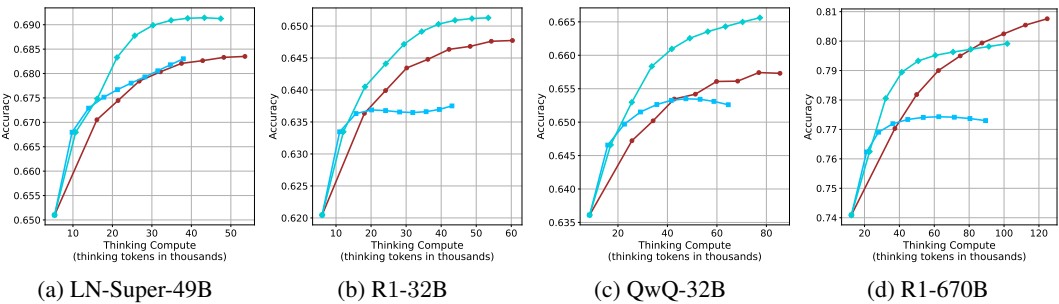

(a) LN-Super-49B    (b) R1-32B    (c) QwQ-32B    (d) R1-670B

Figure 20: Comparing different methods for the GPQA-D benchmark under sequential compute.

# H  SMALL MODELS RESULTS

We present our main results (Sections 3 and 4) using smaller models. We use Llama-3.1-Nemotron-Nano-8B-v1 [LN-Nano-8B; Bercovich et al., 2025] and R1-Distill-Qwen-7B [R1-7B; Guo et al., 2025]. Table 9 (corresponds to Table 1), presents a comparison between shortest/longest/random generation per example for the smaller models. As observed for larger models, using the shortest answer outperforms both random and longest answers across all benchmarks and models.

Table 9: Comparison between taking the shortest/longest/random generation per example.

|  | GPQA-D | | AIME 2024 | | AIME 2025 | | HMMT | | Math Average | |
|---|---|---|---|---|---|---|---|---|---|---|
|  | Thinking Tokens ↓ | Acc. ↑ | Thinking Tokens ↓ | Acc. ↑ | Thinking Tokens ↓ | Acc. ↑ | Thinking Tokens ↓ | Acc. ↑ | Thinking Tokens ↓ | Acc. ↑ |
| **LN-Nano-8B** | | | | | | | | | | |
| random | 7003 | 52.2 | 10380 | 62.1 | 11869 | 46.5 | 12693 | 34.0 | 11647 | 47.5 |
| longest | 10594 (+51%) | 41.4 | 16801 | 40.0 | 17140 | 33.3 | 18516 | 23.3 | 17486 (+50%) | 32.2 |
| shortest | 3937 (−44%) | **55.1** | 6047 | **70.0** | 7127 | **46.7** | 7508 | **50.0** | 6894(−41%) | **55.6** |
| **R1-7B** | | | | | | | | | | |
| random | 7015 | 35.5 | 11538 | 57.8 | 12377 | 42.2 | 14693 | 25.0 | 12869 | 41.7 |
| longest | 11863 (+69%) | 29.8 | 21997 | 26.7 | 21029 | 26.7 | 23899 | 13.3 | 22308 (+73%) | 22.2 |
| shortest | 3438 (−51%) | **46.5** | 5217 | **76.7** | 6409 | **53.3** | 6950 | **43.3** | 6192 (−52%) | **57.8** |

Next, we analyze the performance of *short-m@k* using smaller models (see details in Section 4). Figure 21, Figure 22 and Figure 23 are the sample-size/compute/time-to-answer results for the small models over the math benchmarks, respectively. Figure 24, Figure 25 and Figure 26 are the sample-size/compute/time-to-answer results for the small models for GPQA-D, respectively.

The performance of *short-m@k* using small models remain consistent with those observed in larger ones. *short-1@k* demonstrates a performance advantage over majority voting in low compute regimes, while *short-3@k* dominates it across all compute budgets.

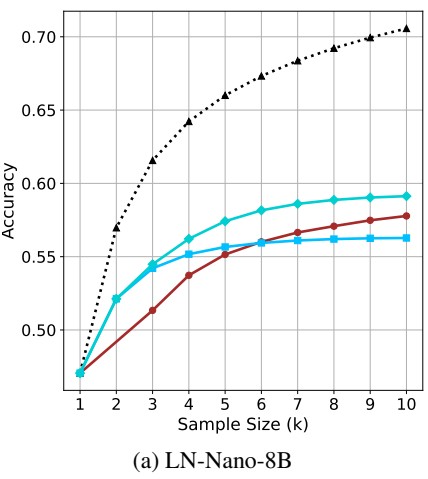
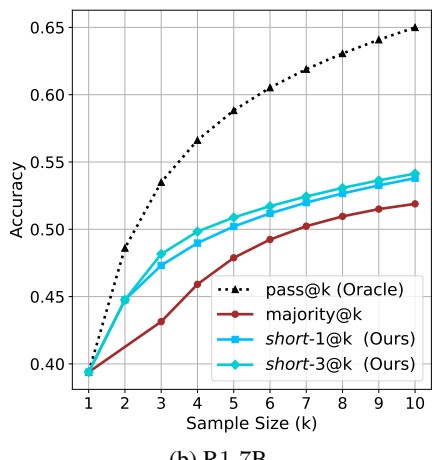

(a) LN-Nano-8B                    (b) R1-7B

Figure 21: Small models - sample size (k) comparison over math benchmarks.

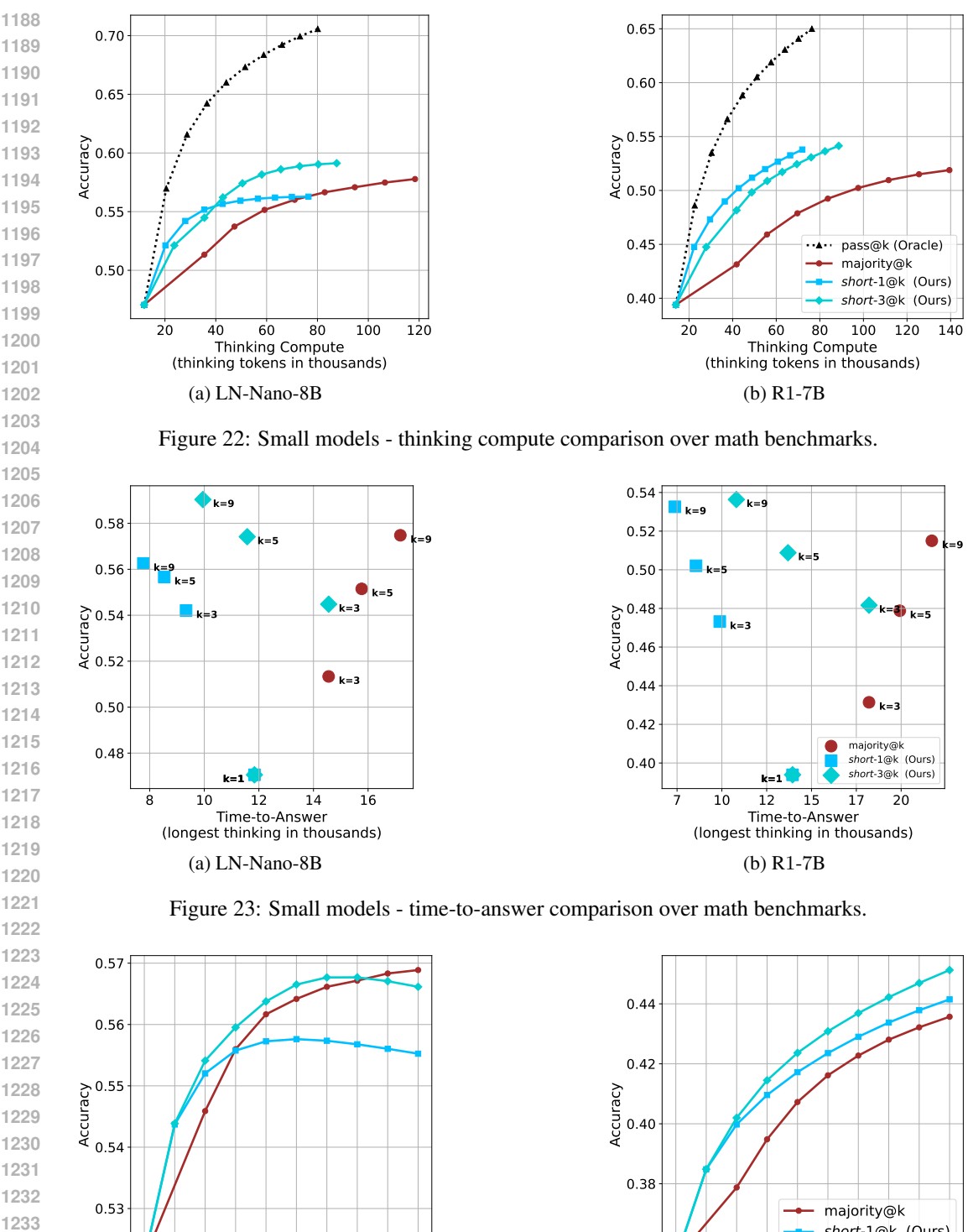

Figure 22: Small models - thinking compute comparison over math benchmarks.

Figure 23: Small models - time-to-answer comparison over math benchmarks.

Figure 24: Small models - sample size ($k$) comparison over GPQA-D.

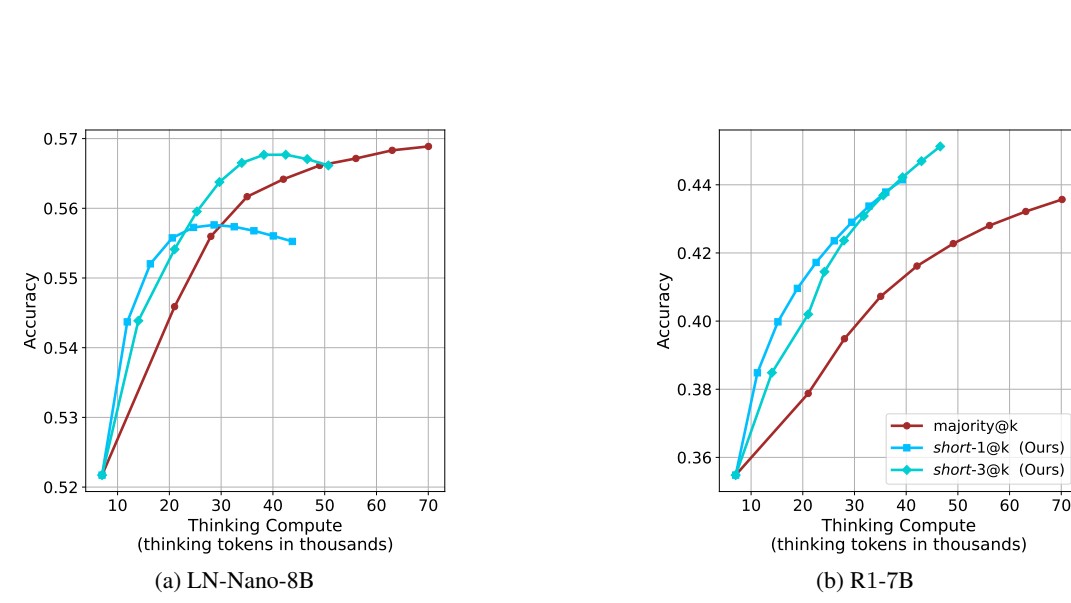

(a) LN-Nano-8B

(b) R1-7B

Figure 25: Small models - thinking compute comparison over GPQA-D.

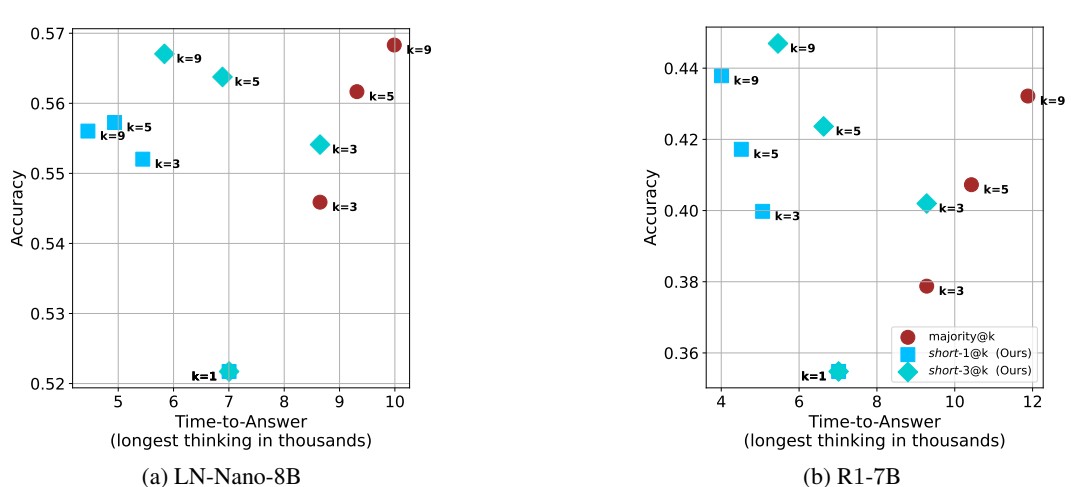

(a) LN-Nano-8B

(b) R1-7B

Figure 26: Small models - time-to-answer comparison over GPQA-D.

