# OpenReview forum: "Don't Overthink it. Preferring Shorter Thinking Chains for Improved LLM Reasoning"
_ICLR.cc/2026/Conference — Submitted to ICLR 2026_

### Official Review · Reviewer_x94b · 2025-10-28

**Soundness:** 3
**Presentation:** 4
**Contribution:** 2
**Rating:** 2
**Confidence:** 5

**Summary:**

This paper challenges the prevailing assumption that longer "thinking chains" and increased test-time compute lead to better LLM reasoning. The authors' central claim is that for a given question, shorter reasoning chains are significantly more likely to yield a correct answer.

Based on this observation, the paper proposes short-m@k, an efficient inference method. This method executes k independent generations in parallel but halts all computation as soon as the first m (fastest) thinking processes are complete. The final answer is then determined by a majority vote among these m short chains.

Through extensive experiments on four models and four reasoning benchmarks, the authors show that short-m@k (particularly short-3@k) consistently outperforms the standard majority@k baseline, achieving higher accuracy while also being substantially faster. The paper also provides an analysis of backtracking behavior and demonstrates that finetuning an LLM on a dataset of short reasoning trajectories (S1-short) improves performance and efficiency over training on long or random trajectories.

**Strengths:**

1. **Practical and Efficient Method:** The short-m@k method is simple to implement, practical, and highly effective. The results showing that short-3@k *simultaneously* improves accuracy and reduces wall-time (up to 33%) over the standard majority@k baseline are significant for practitioners.
2. **Multi-faceted Analysis:** The paper doesn't just present a method; it also provides a multi-dimensional analysis of why it works, including sample size (k), thinking compute (Fig 3), and wall time (Fig 4) is excellent.

**Weaknesses:**

The primary weakness of this paper is the significant lack of novelty. Many of the core findings, while presented as new, closely overlap with recently published work. The paper would be substantially stronger if it properly acknowledged this context and framed its contribution as an extensive validation and practical application of these emerging ideas, rather than as a set of de novo discoveries.

1. **Core finding (shorter is better):** The observation that shorter reasoning chains are more likely to be correct has been independently identified and discussed in prior work, such as [1] and in public discussions around methods like Laconic decoding [3].
2. **The short-m@k method:** This method is presented as a novel contribution. However, short-1@k is functionally equivalent to the "shortest@n" strategy (or Laconic decoding [3]). The short-m@k generalization with an early-exit mechanism is a straightforward and efficient engineering optimization, but not a fundamental  innovation.
3. **Difficulty Analysis:** The finding in Section 5.1 (that harder questions require more thinking, but shorter answers are still preferred within any difficulty bracket) is almost identical to the analysis presented in Figures 3 & 4 of  [1].
4. **Backtrack Analysis:** The analysis in Section 5.2, which finds that correct paths have fewer backtracks, strongly mirrors the concept of "thought-switching" and "under-thinking" from  [1]. Both papers essentially find that incorrect (longer) paths get lost in inefficient, shallow exploration, whereas correct (shorter) paths are more direct and focused.
5. **Finetuning Analysis:** The idea of training models on shorter/preferred trajectories (Section 6) is also not new.  [2] already explored this concept, albeit using DPO instead of SFT. The core principle of preferring shorter, more efficient reasoning paths during training is the same.
6. **Oversimplified Conclusion on Training:** The conclusion that "training on shorter reasoning sequences can lead to better reasoning models" (Section 6) needs to be nuanced. This has a trivial counterexample: training on the absolute shortest sequences (e.g., length 0, direct answers) would cause performance to collapse. The paper does not adequately discuss this trade-off or the risk of  preferring shorter chains.

**References:**

[1] Wang, Yue, et al. "Thoughts are all over the place: On the underthinking of o1-like llms." NeurIPS 2025.

[2] Chen, Xingyu, et al. "Do not think that much for 2+ 3=? on the overthinking of o1-like llms." ICML 2025.

[3] Dimakis, Alex. (2025, Feb 2). X Post. https://x.com/AlexGDimakis/status/1885447830120362099

**Questions:**

My main questions are for clarification on the finetuning experiment in Table 4:

1. **Data Creation:** How were the S1-short/long/random datasets created? The paper states 10 responses were generated per example. Was this simple generation, or was rejection sampling used to ensure the final answer in the trajectory was correct?
2. **Truncation:** What was the maximum generation length used during training and evaluation? Were any samples truncated during training or inference? If so, what was the truncation ratio?

---

> ### Author Response · Authors · 2025-11-21
>
> We thank the reviewer for their valuable review. We appreciate the recognition of our method as "simple to implement, practical, and highly effective," noting its success in "simultaneously improving accuracy and reducing wall-time," and acknowledging our extensive experiments and multi-dimensional analysis.
>
> ---
>
> We acknowledge that the field is rapidly evolving and it is possible we missed some relevant work. We will incorporate the highly recent work  [1,2] into our camera-ready version. Regarding the social media post, we were unaware of it. We agree that many good ideas are being suggested on social media, but we believe that the role of academic research is different – to thoroughly and rigorously study which ideas work and which do not.
>
> We maintain that our core contribution is both important and novel. While previous work has indeed shown that as models think longer on average their performance improves, they lack investigation of what happens if you control for question difficulty. In our work, we show that shorter trajectories outperform long ones within the same question, which to the best of our knowledge no *other work does*. Our further analysis (Section 5.1) resolves this alleged contradiction: indeed, on harder questions, more reasoning is required. Therefore, models that "think longer" can unlock new possibilities of solving such problems, and their overall performance improves. Nonetheless, still, within a specific question (whether easy or hard), shorter answers are far better than longer ones. Again, to the best of our knowledge, *no previous work has provided this insight*. We further show how our insight can be leveraged for getting better and more efficient answers using the short-m@k method.
>
> ---
>
> Regarding our analysis and the fine tuning sections: We agree that the works cited by the reviewer are relevant and confirm the phenomena observed in our fine tuning and analysis sections. We will add those citations and reflect that in our camera ready version. We emphasize that these sections serve primarily as an in-depth analysis and validation of the core findings, rather than the paper's main novel claims. We will also nuance the conclusion of the finetuning section.
>
> ---
>
> Regarding the reviewer questions:
> **Data Creation:** We generated 10 responses per question using the QwQ-32B model (temp=0.7, top_p=0.95). Consistent with the original S1 work [4], no rejection sampling was used during this process.
> **Truncation:** The maximal generation length used consistently throughout the paper was 32,768 tokens, both for inference and training.
>
> We will add these details to the final manuscript.
>
> ---
>
> [4] s1: Simple test-time scaling; Muennighoff et al. ; EMNLP 2025

---

> > ### Comment · Reviewer_x94b · 2025-11-27
> >
> > Thank you for your response.
> >
> > I must admit that the assessment of novelty is somewhat subjective. However, from my perspective, this paper offers very little new knowledge to me. While the authors believe that the core content of this paper is novel, in my view, it seems more like a reinterpretation of existing findings.
> >
> > For instance, the authors state: *"While previous work has indeed shown that as models think longer on average their performance improves, they lack investigation of what happens if you control for question difficulty."* However, in reality, both [1] and [2] include stratified analyses controlling for question difficulty—they just lack the specific conclusion that *"shorter trajectories outperform long ones within the same question."*
> >
> > In my opinion, the significance and novelty of this conclusion feel more like a supplement to existing knowledge, rather than being substantial enough to justify a completely new paper. Therefore, I generally stand by my original assessment.

---

> > > ### Author Response · Authors · 2025-11-27
> > >
> > > We thank the reviewer for their response.
> > >
> > > We agree with the reviewer that assessment of novelty of a given work can often seem subjective. However, we argue that our per-question analysis adds a *significant novel contribution*. To illustrate this, consider Figures 3 & 4 in [1]:
> > >
> > > - Figure 3 shows that harder questions lead to longer generations.
> > > - Figure 4 shows that correct answers lead to shorter generations.
> > >
> > > Combined, one could deduce that the results in Figure 4 are trivial: as models are more successful on easier questions, the proportion of easier questions in the “correct” column is far higher than in “incorrect” one, and therefore the average length of the former is shorter (even for the harder “bucket”).
> > >
> > > Our results clearly demonstrate that **this simple hypothesis is incorrect**. By controlling for the question, we are able to show that correct answers are shorter regardless of question difficulty, and by that shed important light on the behavior of reasoning LLMs.

---

### Official Review · Reviewer_XrSf · 2025-10-31

**Soundness:** 3
**Presentation:** 3
**Contribution:** 2
**Rating:** 4
**Confidence:** 3

**Summary:**

This paper studies the benefit of shorter reasoning chains for LLMs. This contradicts the tendency in some recent papers to extend the reasoning chain to achieve better results.
The first experiment compares chains of different lenths for the same question. Across different (large) models, shorter chains correlate with higher accuracy (picking the shortest chain is better than average and than the longest chain). Based on this observation, the authors propose a simple method to exploit these trends: run k chains in parallel, and once the first m are done, terminate and use majority over those first m. For m=1 and 3, this leads to good results especially with smaller k. With larger k, majority vote sometimes becomes better. The last part is a further evaluation of the short chains. They find that harder questions do need longer chains than simpler questions, in general. Another experiment looks at the frequency and length of backtracking, and finds a correlation between performance and fewer number of backtracks as well as longer backtracks. Finally, they show that finetuning a model on shorter trajectories leads to better outcomes.

**Strengths:**

- the observation that for a single question, shorter chains are better, is interesting. At least in part this seems to stem from fewer backtracks. The observation about training on shorter trajectories I interesting and makes sense, too.

- the proposed method is simple and seems to perform well.

- experiments are performed with very large models.

**Weaknesses:**

- given that other concurrent and prior works have explored the relation between chain length and accuracy, I am wondering what exact insights of this paper go beyond prior / concurrent work. Probably the proposed selection method. Which of the other insights?

- There are a few questions I still have (see below). Based on the answers to those and the above one I am willing to reconsider my score.

**Questions:**

- Sec. 4.3: How were the plots in Fig. 3 generated? Were the models terminated after the specific compute budget? If yes, did you terminate all parallel chains or did you run them in sequence and just didn't do the last ones if budget was over? Or how did you get majority results for a specific budget?

- Sec 5.1: I did not fully understand the setup here. How exactly did you split the questions. By accuracy of one specific model? Or per model? Or average?

- The shorter chain observation is interesting. The experiments are done with fairly large models, I wonder whether the observations also have to do with model size, i.e., the stronger models can do with shorter chains?

- This is what I was also wondering about Section 6. The teacher and student model are fairly similar (same size). Would the short trajectories of the 32B model also work well for architecturally different models?

- sometimes, the 1@k performance goes down as k grows. Does this mean the model generates more wrong short sequences? This seems to happen more for R1-670B and the QwQ-32B model. Does this have something to do with certain properties of the model?

---

> ### Author Response · Authors · 2025-11-21
>
> We thank the reviewer for their review and feedback. We thank them for thinking that our observations (Section 3) and training experiments in (Section 6) are interesting. We are glad that the reviewer thinks that our method is simple and performs well. We also thank them for acknowledging our experiments for using very large and leading models.
>
> ---
>
> **Regarding the insights of the paper, given concurrent work.**
> Previous work has indeed shown that as models think longer on average, their performance improves. This could hint that "longer is better". However, a key missing investigation, which our work adds (and to the best of our knowledge no *other work does*), is what happens if you control for question difficulty. In this case, as we show, shorter trajectories outperform long ones. Our further analysis (Section 5.1) resolves this alleged contradiction: indeed, on harder questions, more reasoning is required. Therefore, models that "think longer" can unlock new possibilities of solving such problems, and their overall performance improves. Nonetheless, still, within a specific question (whether easy or hard), shorter answers are far more accurate than longer ones. Again, to the best of our knowledge, *no previous work has provided this insight*. Our findings lead us to suggest short-m@k, an inference method for reasoning LLMs that provides similar or superior performance compared to majority voting, while far being more efficient.
>
> ---
>
> **Regarding the results presented in figure 3.**
> We clarify that no fixed compute budget was pre-assigned per model. Instead, models generate $k$ trajectories in parallel (limited only by a maximum of 32k generated tokens), and halt the computation according to the inference method examined. For our short-m@k we halt computation for once $m$ thinking trajectories are ended, and for ‘majority’ we wait until all generations fully finished. The reported compute cost is calculated post-hoc based on the total tokens used in these completed runs.
>
> ---
>
> **Regarding the setup in Section 5.1.** We split the questions per model according to the model success rate over each question (based on 20 answers per model per question), meaning the "easy" questions set can differ across models.
>
> ---
>
> **Regarding the questions about model size.**  As per the reviewer request, we ran our experiments on 2 smaller models: Llama-3.1-Nemotron-Nano-8B-v1 (LN-Nano-8B; [1]) and R1-Distill-Qwen-7B (R1-7B; [2]). The results are presented below (correspond to Table 1):
>
> | Setting   | GPQA-D Thinking ↓ | GPQA-D Acc ↑ | AIME 2024 Thinking ↓ | AIME 2024 Acc ↑ | AIME 2025 Thinking ↓ | AIME 2025 Acc ↑ | HMMT Thinking ↓ | HMMT Acc ↑ | Math Avg Thinking ↓ | Math Avg Acc ↑ |
> |---|---:|---:|---:|---:|---:|---:|---:|---:|---:|---:|
> | **LN-Nano-8B** |
> | random   | 7003 | 52 | 10380 | 62 | 11869 | 46 | 12693 | 34 | 11647 | 47.5 |
> | longest  | 10594(+51%) | 41.4 | 16801 | 40.0 | 17140 | 33.3 | 18516 | 23.3 | 17486(+50%) | 32.2 |
> | shortest | 3937(-44%) | **55.1** | 6047 | **70.0** | 7127 | **46.7** | 7508 | **50.0** | 6894(-41%) | **55.6** |
> | **R1-7B** |
> | random   | 7015 | 35.5 | 11538 | 57.8 | 12377 | 42.2 | 14693 | 25.0 | 12869 | 41.7 |
> | longest  | 11863(+69%) | 29.8 | 21997 | 26.7 | 21029 | 26.7 | 23899 | 13.3 | 22308(+73%) | 22.2 |
> | shortest | 3438(-51%) | **46.5** | 5217 | **76.7** | 6409 | **53.3** | 6950 | **43.3** | 6192(-52%) | **57.8** |
>
> As can be seen, the results of the smaller models follow the same trends observed in larger ones.
>
> We also employed our _short-m@k_ method using these small models and found them to have the same trends as their bigger counterparts. We attached those graphs and results into a new appendix in the paper (appendix G).
>
> ---
>
> **Regarding the small decline in performance for short-1@k at large values of $k$.** We acknowledge that as the $k$ increases, the probability of sampling at least one incorrect solution (whether short or long) grows proportionally. While the generation of an incorrect yet short solution remains a low-probability event (as our results suggest), this low-probability event is more likely to be encountered when $k$ is large. Since short-1@k is defined to select the single shortest answer, this mechanism sometimes (for larger $k$s) elevates a short incorrect solution.
>
> We note that **we find it highly encouraging that this performance drop is relatively moderate.** The drop should be interpreted against a baseline: if the selection of a correct or incorrect solution were independent of trajectory length, the short-1@k performance curve would be flat across all $k$. The fact that our curve remains higher than the baseline, especially for smaller $k$, underscores that shorter trajectories are more likely to be correct.
> Crucially, when we evaluate _short-3@k_, the performance decline is not observed at any $k$, while the massive efficiency gain compared to majority voting is preserved.
>
> ---
>
> [1] arxiv.org/abs/2505.00949
>
> [2] arxiv.org/abs/2501.12948

---

> > ### Comment · Reviewer_XrSf · 2025-11-24
> >
> > Thank you for the clarifications. It will be useful to clarify all of these in the paper as well, as other readers may wonder about the same things.
> >
> > I had one more comment about your reply regarding "novelty":
> >
> > "Regarding the insights of the paper, given concurrent work. Previous work has indeed shown that as models think longer on average, their performance improves. This could hint that "longer is better". However, a key missing investigation, which our work adds (and to the best of our knowledge no other work does), is what happens if you control for question difficulty. "
> >
> > This is indeed an interesting question. However, the introduction of the paper seems to focus on "shorter is better" as the main message, the relation to task difficulty is only mentioned in one paragraph in the introduction. This could be re-positioned. Second, there are concurrent works that also notice the dependence on task difficulty, e.g. Wu et al 2025 from the related works section. Could you please clarify the difference to that work?
> > How does the proposed method compare to existing works?

---

> > > ### Author Response · Authors · 2025-11-25
> > >
> > > We thank the reviewer for his comments. We will integrate the clarifications made above to our final version.
> > >
> > > ---
> > >
> > > Regarding the positioning of “shorter is better” at the introduction section. Our work indeed argues that “shorter is better” throughout the paper. What is novel about our work is that we do so even **when task difficulty is controlled**. We do so by investigating each individual question, and measuring performance using short/long generations. We will make sure to clarify this point in the final version.
> > >
> > >
> > > ---
> > >
> > > As per concurrent works that investigates the relation between task difficulty and the reasoning length. We agree that there are several works (such as [Wu et al; 2025]) that state the harder the question - the longer the reasoning needed. This mirrors our results in section 5.1 and the common knowledge that “longer is better”. Having said that, our work focuses on the impact of reasoning length **within difficulty level**, which is controlled by looking over each specific question. Our investigation shows that regardless of task difficulty, shorter trajectories outperform longer ones. This provides a more comprehensive and controlled verification of the claim that “shorter is better”. To the best of our knowledge, this investigation is unique for our work. We will make sure to highlight it in the manuscript.
> > >
> > >
> > > ---
> > >
> > > Additionally, per the reviewer request, we performed an additional fine-tuning experiment using a smaller model, specifically Qwen-2.5-7B-Instruct. We used the same S1 data variants (generated using QwQ-32B) as described in the paper. Results are detailed below:
> > >
> > > | S1 data variant   | GPQA-D Thinking ↓ | GPQA-D Acc ↑ | AIME 2024 Thinking ↓ | AIME 2024 Acc ↑ | AIME 2025 Thinking ↓ | AIME 2025 Acc ↑ | HMMT Thinking ↓ | HMMT Acc ↑ | Math Avg Thinking ↓ | Math Avg Acc ↑ |
> > > |---|---:|---:|---:|---:|---:|---:|---:|---:|---:|---:|
> > > | S1-random (7B)   | 14095 | 39.1 | 25207 | **22.0** | 23822 | **22.0** | 25028 | 10.8 | 24686 | **18.2** |
> > > | S1-long (7B)  | 15528(+10%) | 38.5 | 26210 | 21.7 | 24395 | 19.5 | 26153 | 9.2 | 25586(+3.7%) | 16.8 |
> > > | S1-short (7B) | 13093(-7%) | **40.3** | 24495 | **22.0** | 21945 | 20.8 | 23329 | **11.2** | 23256(-5.8%) | 18.0 |
> > >
> > > While the absolute performance is much lower compared to the 32B model experiment, the trends in terms of trajectories length are similar. S1-short demonstrates shorter thinking trajectories compared to the S1-random baseline (a reduction of $5.8\%$ on mathematics benchmarks and $7\%$ on GPQA-D), while S1-long shows increased trajectories length. Regarding overall performance, consistent with the 32B model findings, S1-short achieved higher accuracy than S1-long. When comparing S1-short to S1-random, it only matches the baseline for the math benchmarks (unlike the 32B experiment), while outperforming it for GPQA-D.
> > >
> > >
> > > ---
> > >
> > > We also note that the appendix regarding small models has moved to appendix H.

---

### Official Review · Reviewer_Qd6G · 2025-11-01

**Soundness:** 3
**Presentation:** 3
**Contribution:** 2
**Rating:** 4
**Confidence:** 4

**Summary:**

This paper studies a counter-intuitive claim: within a single question, shorter thinking chains are more likely to be correct than longer ones. Based on this observation, the authors propose short-m@k, an early-termination inference scheme that stops generation once the first m reasoning trajectories finish. Experiments are fairly comprehensive and indicate that short-m@k yields efficiency benefits while preserving or sometimes improving accuracy.

**Strengths:**

- The proposed short-m@k inference scheme sounds like a pragmatic knob to trade inference time for accuracy, and the compute/time reduction angle is appealing.
- The empirical evaluation is broad: multiple reasoning LLMs, multiple math datasets, compute/time/accuracy slicing.

**Weaknesses:**

**W1. Prior work draws almost the opposite conclusion.** For example,  [1] explicitly encourages longer trajectories and then uses self-consistency voting over longer chains. This paper reports the opposite monotonic trend. The authors do not directly reconcile this contradiction. Clarifying this contradiction is necessary before readers can interpret the result as a generally valid principle.

**W2. The novelty is unclear.** The related work section itself (line ~141 *More relevant to our work…*) already cites multiple recent works that are extremely close in core conclusion. Several of these already suggest “shorter is often better”. I am not convinced what new conceptual contribution this paper adds beyond bundling those observations and re-running them under a unified evaluation protocol. The paper needs a much sharper novelty claim and direct head-to-head comparison to the most similar baselines.

[1] “Chain-of-Thought Reasoning Without Prompting”, NeurIPS 2024, Wang et al.

**Questions:**

see weakness

---

> ### Author Response · Authors · 2025-11-21
>
> We thank the reviewer for their feedback. We were pleased that they found our method's ability to yield efficiency while preserving or improving performance, and that they found our experiments to be comprehensive and broad.
>
> ---
>
> **Regarding relation to prior and concurrent work.**
>
> Previous work has indeed shown that as models think longer on average, their performance improves. This could hint that "longer is better". However, a key missing investigation, which our work adds (and to the best of our knowledge no *other work does*), is what happens if you control for question difficulty. In this case, as we show, shorter trajectories outperform long ones. Our further analysis (Section 5.1) resolves this alleged contradiction: indeed, on harder questions, more reasoning is required. Therefore, models that "think longer" can unlock new possibilities of solving such problems, and their overall performance improves. Nonetheless, still, within a specific question (whether easy or hard), shorter answers are far more accurate than longer ones. Again, to the best of our knowledge, *no previous work has provided this insight*. Our findings lead us to suggest short-m@k, an inference method for reasoning LLMs that provides similar or superior performance compared to majority voting, while far being more efficient.
>
> Per the reviewer specific reference [1], we note that this paper was published before the reasoning LLMs era, and as such it does not use explicit thinking chains (which includes in depth thinking, including back-and-fourth and such), but only direct chain-of-thoughts. As such, it is not trivial to compare their findings to our examined models.

---

> > ### Comment · Reviewer_Qd6G · 2025-11-27
> >
> > I apologize for the earlier incorrect citation. The work I intended to reference is “Complexity-Based Prompting for Multi-Step Reasoning” (Fu et al., 2023). In that paper, the authors rely on self-consistency voting over longer CoT chains, and the proposed method in your submission appears closely related. Your approach mainly seems to **switch the filtering strategy from short-filtered to long-filtered chains derived from the same SC-style sampling**. Also, I notice the author didn't reply to W2. Since my previous reference was wrong, I would like to give the authors the opportunity to respond again to W2 on **new conceptual contribution this paper adds beyond the related works**.

---

> > > ### Author Response · Authors · 2025-11-27
> > >
> > > We thank the reviewer for their response.
> > >
> > > Regarding the study suggested by Fu et al., 2023. Their main finding is that long and complex CoT correspond to better performance of non-reasoning LLMs. In our work we show for reasoning LLMS the opposite holds true: shorter reasoning trajectories are better. We agree that our proposed method is similar to the proposed method of Fu et al., 2023, however our method critically selects the shortest reasoning answers, not the longest ones. This is a non-trivial difference for the use of reasoning LLMs, resulting in superior performance while also yielding substantial compute and time savings.
> > >
> > > As for W2, we acknowledge the rapidly emerging nature of the reasoning LLM field. In our work, we shed light over the relation between reasoning length and performance **within the same question**, showing that shorter reasoning chains yield superior results. This approach is crucial as it avoids the potential bias inherent in aggregating results across questions, where models might simply succeed on easier ones that intrinsically require shorter chains (see Table 2). We also note that based on our insights, we provide a new inference method _short-m@k_ which achieves inference time efficiency while consistently matching or exceeding the performance of majority voting.

---

### Official Review · Reviewer_W8q7 · 2025-11-02

**Soundness:** 2
**Presentation:** 3
**Contribution:** 3
**Rating:** 8
**Confidence:** 4

**Summary:**

The paper challenges the view that longer reasoning chains improve LLM performance, which is a widely held assumption in reasoning LLM research. The authors analyze multiple reasoning benchmarks using leading models and find that shorter reasoning trajectories are significantly more likely to yield correct answers. Based on these findings, they propose short-m@k, an inference method that executes k parallel generations and halts once the first m finish, selecting answers by majority vote. Experiments show that short-1@k matches or exceeds majority voting while reducing compute by up to 40%, and short-3@k improves accuracy. Finetuning on shorter reasoning chains further improves performance and efficiency.

**Strengths:**

* The study focuses on an important topic in reasoning LLMs by challenging that longer chains enhance reasoning.
* The proposed short-m@k framework introduces an elegant parallel decoding mechanism. This approach is well-motivated by the authors’ empirical findings and leads to measurable compute and time savings.
* The authors validate across four major reasoning models and multiple benchmarks, combining performance, compute, and wall-time analyses. Additional fine-tuning experiments support the generality of their claims.

**Weaknesses:**

* The paper primarily provides empirical evidence without formal analysis of why shorter chains outperform longer ones. Also see questions.
* The proposed method assumes access to batch inference resources; its effectiveness under memory-constrained or latency-constrained conditions remains unclear. Evaluation in sequential or streaming inference settings could provide further robustness evidence.

**Questions:**

* Several previous works claim that a medium CoT length would achieve the best performance [1], instead of the shortest one as stated in the paper. The authors may conduct more experiments using some smaller models or easier datasets to determine whether the phenomenon only exists in complex tasks or large models.
* How sensitive is short-m@k to the choice of m and k across different task complexities? Would dynamic selection during inference outperform static hyperparameters?
* Does fine-tuning on short reasoning chains risk reducing a model’s capacity to handle genuinely long reasoning problems (such as tasks that require long-context modeling)? Are there observable trade-offs in general reasoning robustness?

---

> ### Author Response · Authors · 2025-11-21
>
> We thank the reviewer for their positive feedback. We are pleased that they found our study to focus on an **important topic in reasoning LLMs**, and we appreciate them characterizing our method as **elegant** and leading to **measurable compute and time savings**. We also thank the reviewer for acknowledging our experimental setup.
>
> ---
>
> **Regarding further understanding of why shorter trajectories are better.**
> We appreciate the reviewer raising this important point. To better understand the mechanism behind the preference for shorter thinking, we conducted a backtrack analysis of the trajectories, as described in Section 5.2. We find out that **correct trajectories consistently exhibit fewer backtracks** (which correspond to shorter trajectories) compared to incorrect ones. Moreover, we find that **correct trajectories have longer back track length**, which suggests a more in-depth step. We next turn to analyze the trajectories in a length controlled manner, and find that within the same length - **correct trajectories use fewer backtracks**, which suggests a more direct way to the answer. While this analysis does not provide a full explanation to our observed results, it does suggest that shorter trajectories correspond to a more in-depth path to the answer, with fewer backtracks.
>
> ---
>
> **Regarding batch inference.**
> As noted in the limitations section, our short-m@k method would lead to best compute savings when batch decoding is used. Nevertheless, short-m@k can also be used without batch decoding and lead to efficiency gains, as even in this setup, any generation that runs longer than the current m shortest runs can be terminated. For instance, in short-1@k, on average the shortest run will be the k/2 run, so any run afterwards can be terminated. In fact, earlier generations can also be terminated earlier, assuming they are longer than the shortest one thus far, leading to additional gains. We will add an analysis of potential savings as a function of the batch size to the next version.
>
> ---
>
> **Regarding using smaller models.**
> As per the reviewer request, we ran experiments on 2 smaller models: Llama-3.1-Nemotron-Nano-8B-v1 (LN-Nano-8B; [1]) and R1-Distill-Qwen-7B (R1-7B; [2]). The results, which correspond to Table 1 in the paper, are presented below:
>
> | Setting   | GPQA-D Thinking ↓ | GPQA-D Acc ↑ | AIME 2024 Thinking ↓ | AIME 2024 Acc ↑ | AIME 2025 Thinking ↓ | AIME 2025 Acc ↑ | HMMT Thinking ↓ | HMMT Acc ↑ | Math Avg Thinking ↓ | Math Avg Acc ↑ |
> |---|---:|---:|---:|---:|---:|---:|---:|---:|---:|---:|
> | **LN-Nano-8B** |
> | random   | 7003 | 52 | 10380 | 62 | 11869 | 46 | 12693 | 34 | 11647 | 47.5 |
> | longest  | 10594(+51%) | 41.4 | 16801 | 40.0 | 17140 | 33.3 | 18516 | 23.3 | 17486(+50%) | 32.2 |
> | shortest | 3937(-44%) | **55.1** | 6047 | **70.0** | 7127 | **46.7** | 7508 | **50.0** | 6894(-41%) | **55.6** |
> | **R1-7B** |
> | random   | 7015 | 35.5 | 11538 | 57.8 | 12377 | 42.2 | 14693 | 25.0 | 12869 | 41.7 |
> | longest  | 11863(+69%) | 29.8 | 21997 | 26.7 | 21029 | 26.7 | 23899 | 13.3 | 22308(+73%) | 22.2 |
> | shortest | 3438(-51%) | **46.5** | 5217 | **76.7** | 6409 | **53.3** | 6950 | **43.3** | 6192(-52%) | **57.8** |
>
> As can be seen, the results of the smaller models follow the same trends observed for larger ones.
>
> We also employed the _short-m@k_ method using these smaller models and found them to have the same trends as their bigger counterparts. We attached those graphs and results into a new appendix in the paper (appendix G).
>
> ---
>
> **Regarding the choice of m and k.**
>  We thank the reviewer for this comment. We first note that for short-3@k it seems that the larger the k the better, so one should allocate as large k as the compute budget allows (as batched inference requires more memory).
>
>
> As for the choice of m, we point to figure 17a (Appendix D) in our paper, which performs an ablation study over m. As can be seen, larger m values seem to perform similarly. Thus, our recommendation is to use m=3.
>
> We agree that a dynamic allocation of resources ($k$ and $m$) based on question difficulty is an interesting direction, however we believe this is out of scope for this submission, hence we leave it for future research.
>
> ---
>
> **Regarding the finetuning experiment.**
> Regarding hurting performance of the model over long reasoning problems, we point out that the S1-short variant of the data is not that short. The length distribution, as presented in Figure 18(a), is shifted toward shorter chains but still includes trajectories up to 30k tokens in length. This is comparable to the length range found in the base S1-random dataset. The distribution itself is shifted towards shorter chains, but as can be seen, it is not short in the sense that the model can forget his long reasoning capabilities.
>
> Additionally, we did not observe any specific, major trade-offs in general reasoning robustness during our experiments.
>
> ---
>
> [1] arxiv.org/abs/2505.00949
>
> [2] arxiv.org/abs/2501.12948

---

> > ### Author Response · Authors · 2025-11-25
> >
> > Based on the reviewer's suggestion and our above response, we evaluate the performance of our proposed short-m@k method under a sequential (instead of batched decoding). We’ve added the results into a new appendix: “Sequential Results” (currently appendix G).
> >
> > As can be seen, while a sequential setup inherently lowers the compute benefits of batched decoding, the overall computational efficiency of short-m@k compared to majority voting remains. Although the total count of intermediate "thinking tokens" for short-m@k (with m={1, 3}) is larger than in the batched decoding setting, the compute saving remains substantial (up to 40% compute saving for the same accuracy). We can still observe better performance for short-1@k and short-3@k when considering low compute budgets, while for larger ones short-3@k performs similarly or even better compared to majority voting.
> >
> > We thank the reviewer for helping us strengthen the paper, and we will integrate this comprehensive sequential analysis into the final manuscript.
> >
> > ---
> >
> > We also note that due to the addition of that appendix, the appendix regarding small models has moved to appendix H.

---

### Official Review · Reviewer_Qq2x · 2025-11-11

**Soundness:** 3
**Presentation:** 3
**Contribution:** 2
**Rating:** 4
**Confidence:** 3

**Summary:**

This paper proposes a simple method short-m@k to select relatively short reasoning chains from LLM generations, and show that even with short-1 and short-3, the accuracies can surpass those from majority@k. The authors have also fine-tuned an LLM using the reasoning chains of different lengths, and it turned out that the short data (S1-short) leads to the best performance.

**Strengths:**

-	The experiments are based on the most difficult challenging task set, such as AIME 2025, HMMT etc., which represents the frontier of LLM reasoning performances.
-	It seems easy to implement the proposed methods and to replicate the experiments on other models.
-	The topic is a core issue faced by most reasoning LLMs.

**Weaknesses:**

-	In general, the findings of this paper are a bit empirical, which lacks theoretical insights, or interpretations from case-by-case analysis, about *why* shorter reasoning trajectories are more beneficial than longer ones. Also, questions like *how much longer* the thinking process should be for harder tasks can be asked.
-	The S1 data seems very an important one to validate the results, but its nature and how it is constructed seems not introduced as all. I understood that this is from other folks’ work, but giving more context to your method would really encourage readers outside the field. (I did not realize the existence of works like S1 until its recent publication, and TBH, it is not realistic for all reviewers to carefully track down the origins of any cited dataset through bibliography)
-	As the authors have pointed out, the fine-tuning experiments are limited to only one model on one task.

**Questions:**

Can the authors provide more details about the “batch decoding” mentioned at line 650, as it seems particularly important to your implementation? Thanks.

---

> ### Author Response · Authors · 2025-11-21
>
> We thank the reviewer for their comprehensive feedback. We are pleased that the reviewer recognizes the relevance and importance of our paper's topic for reasoning LLMs, and acknowledges the simplicity of implementation and the challenging nature of the benchmarks used to validate our insights.
>
> ---
>
> **Regarding why shorter thinking is preferable.**
> We appreciate the reviewer raising this important point. To better understand the mechanism behind the preference for shorter thinking, we conducted a backtrack analysis of the trajectories, as described in Section 5.2. We find out that **correct trajectories consistently exhibit fewer backtracks** (which correspond to shorter trajectories) compared to incorrect ones. Moreover, we find that **correct trajectories have longer back track length**, which suggests a more in-depth step. We next turn to analyze the trajectories in a length controlled manner, and find that within the same length - **correct trajectories use fewer backtracks**, which suggests a more direct way to the answer. While this analysis does not provide a full explanation to our observed results, it does suggest that shorter trajectories correspond to a more in-depth path to the answer, with fewer backtracks.
>
> We agree that the question of "how much longer the thinking process should be for harder tasks" is insightful. While the term "hard questions" is subjective, we define it in Section 5.1 as those where a specific model struggles to produce the correct answer. Under that definition, we do see that models use a higher number of reasoning tokens for harder questions; nevertheless, the amount of thinking tokens used is still model-dependent. Lastly, determining the optimal token allocation a priori to optimize reasoning is an interesting research direction, but we believe this is highly non-trivial (maybe as hard as solving the question itself), and thus falls out of the scope of the work.
>
> ---
>
> **Regarding the finetuning experiment.** We thank the reviewer for pointing out the missing information about this experiment. The S1 finetuning paradigm converts non-reasoning models to reasoning ones with a simple SFT over a small amount of distilled data [1]. Specifically, the authors suggest using 1k selected examples to incent the reasoning behavior. The examples span the following topics Astronomy, Biology, Chemistry, Computer Science, Geography, Mathematics, English, Law, Logic and Physics. The selection of the questions is based on quality, difficulty and diversity (more details and deflations on those can be found in their paper). Given those 1k questions, a reasoning model (QwQ-32B in our case) is used to answer them, and the reasoning trajectories are used to train a non-reasoning model “how to think”.
> As for the fine-tuned model, we note that this experiment is only meant to validate our main findings (shorter trajectories lead to better performance). As such, we strictly follow the S1 paper, which experimented with a single model –  Qwen-2.5-32B-Instruct.
>
> As for the evaluation tasks, we measure our finetuned models over **4** tasks rather on 1 (as done throughout the paper). Regarding the model selection for the finetuning, we follow the same recipe as suggested by the S1 authors.
>
> ---
> As for batch decoding, we clarify that by "batch decoding" we mean using the batch dimension of the LLM to generate multiple independent answers in parallel for a single question. Specifically, in a scenario where $k=4$, the batch dimension of the model is used in order to generate 4 independent answers for the same question in parallel. This approach is leveraged to maximize hardware utilization and efficiency during the search process.
> We will add all the missing info to the final manuscript.
>
> [1] s1: Simple test-time scaling; Muennighoff et al. ; EMNLP 2025

---

> > ### Author Response · Authors · 2025-11-25
> >
> > Per the reviewer request, we conducted an additional fine-tuning experiment with a smaller model. We used Qwen-2.5-7B-Instruct as our base model, using the same S1 data variants presented in the paper (which are based on QwQ-32B). Results are presented below:
> >
> > | S1 data variant   | GPQA-D Thinking ↓ | GPQA-D Acc ↑ | AIME 2024 Thinking ↓ | AIME 2024 Acc ↑ | AIME 2025 Thinking ↓ | AIME 2025 Acc ↑ | HMMT Thinking ↓ | HMMT Acc ↑ | Math Avg Thinking ↓ | Math Avg Acc ↑ |
> > |---|---:|---:|---:|---:|---:|---:|---:|---:|---:|---:|
> > | S1-random (7B)   | 14095 | 39.1 | 25207 | **22.0** | 23822 | **22.0** | 25028 | 10.8 | 24686 | **18.2** |
> > | S1-long (7B)  | 15528(+10%) | 38.5 | 26210 | 21.7 | 24395 | 19.5 | 26153 | 9.2 | 25586(+3.7%) | 16.8 |
> > | S1-short (7B) | 13093(-7%) | **40.3** | 24495 | **22.0** | 21945 | 20.8 | 23329 | **11.2** | 23256(-5.8%) | 18.0 |
> >
> > While the absolute performance is much lower compared to the 32B model experiment, the trends in terms of trajectories length are similar. S1-short demonstrates shorter thinking trajectories compared to the S1-random baseline (a reduction of $5.8\%$ on mathematics benchmarks and $7\%$ on GPQA-D), while S1-long shows increased trajectories length. Regarding overall performance, consistent with the 32B model findings, S1-short achieved higher accuracy than S1-long. When comparing S1-short to S1-random, it only matches the baseline for the math benchmarks (unlike the 32B experiment), while outperforming it for GPQA-D.

---

### Author Response · Authors · 2025-11-21

We thank all reviewers for their comprehensive and valuable feedback. We are pleased that Reviewers Qq2x and W8q7 recognize the relevance and importance of our paper's topic for reasoning LLMs. Our method was consistently framed as being simple, practical, and elegant (Qq2x, W8q7, XrSf, x94b), and we are delighted that it was found to yield efficiency while preserving or improving performance (W8q7, Qd6G, XrSf). We thank Reviewers Qq2x, Qd6G and x94b for acknowledging the comprehensive and extensive nature of our experiments.


We have addressed each reviewer's specific concerns point-by-point in their respective threads.

---

### Author Response · Authors · 2025-11-30
**Final Remarks**

We thank the reviewers and the AC for their time and constructive feedback, which improved our paper. We are pleased that the reviewers recognized the relevance and importance of our work (Qq2x and W8q7), praised our method as simple, practical, and elegant, and noted its efficiency and strong performance (Qq2x, W8q7, XrSf, x94b and Qd6G). We also appreciate the acknowledgment of our comprehensive and extensive experiments (Qq2x, Qd6G and x94b).

We wish to briefly summarize the main points raised during the author-reviewer discussion. We are happy to further clarify any additional points.

---

**Regarding the novelty concern.**

Our core finding is that the shorter reasoning is preferable for reasoning LLMs **within the same question** (which controls for difficulty). This in contrast to previous works that has shown that as models think longer on average, their performance improves, which could hint that longer reasoning is better. We emphasize that a key missing investigation, which our work adds (and to the best of our knowledge no other work does), is what happens if you control for question difficulty. To fully control the difficulty we examine each question individually and find that shorter trajectories outperform long ones. Our further analysis resolves this alleged contradiction: indeed, on harder questions, more reasoning is required, nonetheless, within a specific question (whether easy or hard), shorter answers are far more accurate than longer ones.

Our findings lead us to suggest _short-m@k_, an inference method for reasoning LLMs that provides similar or superior performance compared to majority voting, while being far more efficient.

---

**Regarding adding smaller models.**

We’ve added two new models (Llama-3.1-Nemotron-Nano-8B-v1 and R1-Distill-Qwen-7B) in order to understand how shorter reasoning behaves for smaller models. Our new results suggest that smaller models follow the same trends observed for larger ones. We also employed the _short-m@k_ method using these smaller models and found them to have the same trends as their bigger counterparts. We attached those graphs and results into a new appendix in the paper (appendix H).

We’ve also conducted an additional fine-tuning experiment with a smaller model (Section 6) using the Qwen-2.5-7B-Instruct as our base model. Our results show similar trends in terms of trajectories length and performance compared to the original experiment with a larger base model. We will add those results into our final version.

---

**Regarding non-batched results for _short-m@k_.**

We’ve evaluated the performance _short-m@k_ method under a sequential decoding (instead of batched decoding), results are in appendix G. Our results suggest that while a sequential setup inherently lowers the compute benefits of batched decoding, the overall computational efficiency of _short-m@k_ compared to majority voting remains quite significant (up to 40% reduction in latency).

---

### Meta-Review · Area_Chair_vThZ · 2026-01-02

**Summary:**

This submission argues that within a fixed question, shorter reasoning chains tend to be more accurate than longer ones, and it proposes short‑m@k, an inference strategy that runs k parallel generations but stops once the first m finish and then majority-votes over those shortest completions. Reviewers broadly agreed the empirical phenomenon is interesting and the proposed inference rule is practical.

The main concerns raised across reviews were: differentiation from closely related work, being primarily empirical, practicality beyond parallel decoding. The rebuttal and discussion meaningfully strengthened the paper on practicality and clarity. However, the novelty concern remains the primary unresolved issue, with at least one reviewer likely explicitly maintaining a reject position after discussion. Overall, the post-rebuttal state remains borderline, and my recommendation is driven chiefly by the unresolved differentiation concerns.

**Reviewer Concerns:**

**Reviewer Qq2x (Rating 4: weak reject)**

Concern(s):

Largely empirical; asked for deeper explanation/interpretation of why shorter trajectories help and how thinking length should scale with difficulty.

Insufficient context on the S1 paradigm and how the fine-tuning dataset is constructed.

Fine-tuning validation limited (single model/task in the original framing).

Clarification of what “batch decoding” means operationally.

Addressed by rebuttal / discussion:

Authors pointed to their backtrack analysis as an explanatory lens (fewer backtracks for correct answers; controlled-length analysis), and clarified their definition of “hard” questions via per-model success rate splits.

Added substantially more explanation of S1 and clarified the fine-tuning construction process (10 sampled responses per prompt; selection of short/long/random; no rejection sampling).

Clarified batch decoding as using the batch dimension to generate multiple independent answers for the same query in parallel.

Still outstanding:

The explanation remains descriptive, it helps interpret behavior but does not fully answer “why”.

Fine-tuning generality is improved but still limited in breadth (few base models and a single paradigm family).

**Reviewer W8q7 (Rating 8: accept)**

Concern(s):

Empirical evidence is strong but lacks formal analysis.

Dependence on batch inference; requested evaluation under sequential/streaming settings.

Relationship to prior works suggesting “medium length” is optimal.

Sensitivity of hyperparameters (m, k) and whether dynamic selection might help.

Potential trade-offs of fine-tuning on shorter chains for genuinely long reasoning problems.

Addressed by rebuttal / discussion:

Provided additional analysis and emphasized backtracking results as partial mechanism.

Added sequential decoding evaluation for short‑m@k showing the method retains benefits, though reduced relative to fully batched decoding.

Added small model experiments (8B and 7B) indicating similar qualitative trends.

Still outstanding:

No direct head-to-head evaluation against “optimal/medium-length” selection rules beyond the paper’s proposed scheme and majority voting, so broader claims about optimality across task regimes remain only partially supported.

**Reviewer Qd6G (Rating 4: weak reject)**

Concern(s):

Prior work often suggests longer trajectories (and self-consistency over longer CoT) improve results.

Core claims and parts of the method appear close to multiple recent works; requested sharper novelty statement and comparisons to closest baselines.

Addressed by rebuttal / discussion:

Authors clarified that overall longer thinking can correlate with harder questions, but within a fixed question shorter trajectories tend to be more correct; they emphasize per-question control as the key methodological difference.

Authors acknowledged rapid emergence of related work and tried to reposition novelty around the per-question analysis and the short‑m@k early-termination procedure.

Still outstanding:

The novelty concern remains substantially unresolved: the rebuttal explains what they believe is new, but does not fully resolve whether this is sufficiently distinct from concurrent work in terms of conceptual contribution and baseline comparisons.

**Reviewer XrSf (Rating 4: weak reject)**

Concern(s):

Clarification of how compute-budget curves were generated and how majority results are computed under compute/time constraints.

Clarification of difficulty splits in Section 5.1.

Generality: whether effects depend on model size; whether training transfer holds beyond similar teacher/student settings.

Why short‑1@k can degrade with larger k.

Addressed by rebuttal / discussion:

Authors clarified compute/time were measured post-hoc under each inference rule rather than enforcing a fixed compute budget mid-generation; clarified per-model question splitting by success rates.

Added small model results supporting similar trends across sizes.

Explained short‑1@k degradation with large k as increased chance of sampling a very short but wrong solution; argued short‑3@k mitigates this.

Follow-up discussion clarified intended novelty positioning relative to difficulty-dependent observations in concurrent work.

Still outstanding:

Remaining concern overlaps with broader panel concern: the incremental/novelty question versus concurrent analyses, and lack of direct head-to-head comparisons to the closest prior methods.

**Reviewer x94b (Rating 2: reject)**

Concern(s):

Asserts the core phenomenon and analyses mirror recent papers and public discussions; argues short‑1@k is essentially “shortest-of-n/laconic decoding,” short‑m@k is a straightforward engineering extension, and difficulty/backtrack/fine-tuning analyses are conceptually overlapping with prior work.

Requested details about fine-tuning data creation (e.g., rejection sampling) and truncation.

Addressed by rebuttal / discussion:

Authors acknowledged the need to cite recent papers and promised improved framing/citations in the camera-ready.

Clarified data creation: 10 responses per example; no rejection sampling; maximum generation length consistently 32,768 tokens.

Agreed to nuance the fine-tuning conclusion.

Still outstanding:

The reviewer explicitly maintained their reject stance after discussion, stating that even with the per-question framing the contribution appears supplementary rather than substantial. This is the most significant remaining obstacle to consensus.

**Response to AC comments on novelty**
"Length-accuract relationship within a specific question" is an interesting new direction to study. However, AC echoes the concerns raised by reviewers on (1) practicality beyond batched decoding: unlike test-time scaling that brings better top-N that can be used to improve model via RL, the implication is rather limited; (2) whether the conclusion is universally correct can be better supported, given there exist seemingly conflict observations and lack of theoretically support

**Reviewer Scores:**

Reviewer Qq2x (initial 4 → expected 4)
The rebuttal addressed clarity issues (S1 context, batch decoding) and added supportive evidence, but the main critique (“mostly empirical, limited mechanistic insight; limited fine-tuning scope”) is only partially addressed.

Reviewer W8q7 (initial 8 → expected 8)
This reviewer was already positive. Added sequential/small-model evidence and ablation discussion further supports their accept stance, but likely does not change the score.

Reviewer Qd6G (initial 4 → expected 4)
The rebuttal provides a reconciliation narrative (difficulty vs within-question), but the reviewer’s central concern is novelty and closeness to related work.

Reviewer XrSf (initial 4 → expected 4 or 6, likely 6 if convinced by clarifications)
Many of the reviewer’s concrete methodological questions were answered (compute curves, difficulty split) and additional small-model experiments were provided.

Reviewer x94b (initial 2 → expected 2)
The reviewer doubled down after discussion that novelty remains insufficient; thus no score increase is expected.

---

### Decision · Program_Chairs · 2026-01-26

Reject